# Biomolecular condensates modulate membrane lipid packing and hydration

Agustín Mangiarotti [1] ✉, Macarena Siri [1], Nicky W. Tam[1], Ziliang Zhao [1,4,5], Leonel Malacrida [2,3] ✉ & Rumiana Dimova [1] ✉

Membrane wetting by biomolecular condensates recently emerged as a key phenomenon in cell biology, playing an important role in a diverse range of processes across different organisms. However, an understanding of the molecular mechanisms behind condensate formation and interaction with lipid membranes is still missing. To study this, we exploited the properties of the dyes ACDAN and LAURDAN as nano-environmental sensors in combination with phasor analysis of hyperspectral and lifetime imaging microscopy. Using glycinin as a model condensate-forming protein and giant vesicles as model membranes, we obtained vital information on the process of condensate formation and membrane wetting. Our results reveal that glycinin condensates display differences in water dynamics when changing the salinity of the medium as a consequence of rearrangements in the secondary structure of the protein. Remarkably, analysis of membrane-condensates interaction with protein as well as polymer condensates indicated a correlation between increased wetting affinity and enhanced lipid packing. This is demonstrated by a decrease in the dipolar relaxation of water across all membrane-condensate systems, suggesting a general mechanism to tune membrane packing by condensate wetting.

Biomolecular condensates play a crucial role in cellular organization and metabolism[1]. These dynamic assemblies arise by liquid-liquid phase separation and are involved in a broad range of cell processes, from genomic organization[2,3], to stress responses[4,5], and virus infection[6], to name a few. More recently, the interaction of these membraneless-organelles with membrane-bound compartments has emerged as a new means for intracellular compartmentation, protein assembly, and signaling[7,8]. Among others, membrane-biomolecular condensates interactions are involved in developing tight junctions[9], formation of the autophagosome[10], signal transduction in T cells[11], assembling of virus capsid proteins[12], and in the biogenesis and fission of droplets in the endoplasmic reticulum[13]. Recently, membranes have been reported to control the size of intracellular condensates and

modify their material properties[14]. Furthermore, the crosstalk between membranes and condensates can promote phase separation coupling in the lipid and the protein phases[15–18] also shown in cytoplasm mimetic systems[19,20]. While the field of membrane-condensate interactions is rapidly gaining momentum, important cues are still missing in our understanding of the underlying mechanisms associated with the resulting structural changes and remodeling. Given the small size of condensates, quantifications of these interactions are often precluded in vivo[21]. Thus, critical insight about the material properties of condensates have been obtained in vitro, taking advantage of their relatively ease of their reconstitution[22].

Membrane wetting by condensates was first reported more than a decade ago for giant unilamellar vesicles (GUVs) enclosing an aqueous

[1]Max Planck Institute of Colloids and Interfaces, Science Park Golm, 14476 Potsdam, Germany. [2]Departamento de Fisiopatología, Hospital de Clínicas, Facultad de Medicina, Universidad de la República, Montevideo, Uruguay. [3]Advanced Bioimaging Unit, Institut Pasteur of Montevideo and Universidad de la República, Montevideo, Uruguay. [4]Present address: Leibniz Institute of Photonic Technology e.V., Albert-Einstein-Straße 9, 07745 Jena, Germany. [5]Present address: Institute of Applied Optics and Biophysics, Friedrich-Schiller-University Jena, Max-Wien Platz 1, 07743 Jena, Germany. ✉e-mail: agustin.mangiarotti@mpikg.mpg.de; lmalacrida@pasteur.edu.uy; Rumiana.Dimova@mpikg.mpg.de

two-phase system (ATPS)[19,23]. The ATPS formed from solutions of poly(ethylene glycol) (PEG) and dextran is probably the best understood example. The PEG/dextran ATPS constitutes a convenient model of the crowded cytoplasm to study liquid-liquid phase separation in contact with membranes[24–26]. These pioneering studies revealed that membrane wetting by condensates can give rise to several biologically relevant processes of membrane remodeling, such as inward and outward budding, nanotube formation, and fission of membrane compartments[24]. GUV tubulation was also observed when condensate-forming protein species are bound to the membrane via NTA-lipids[27]. Recently, we addressed the interaction between non-anchored protein condensates and membranes, the different morphological changes and wetting transitions driven by this interaction and provided a theoretical framework that allows quantitative analysis[28]. We showed that the interaction between condensates and membranes can lead to the complex mutual remodeling of the membrane-condensate interface, producing microscopic membranous protrusions[28]. These observations constituted the first steps in understanding the interaction between membranes and condensates at the micron-scale, but the underlying molecular mechanisms remain elusive and largely unexplored.

The protein we use in this work to study phase separation and membrane wetting, glycinin, is one of the most abundant storage proteins in the soybean. Glycinin, along with other storage proteins, plays a crucial role in promoting vacuole membrane remodeling during embryogenesis in plants[28–30]. Recent in vitro experiments have demonstrated its potential of membrane remodeling, highlighting its efficacy as a robust model for studying membrane-condensate interactions[28]. Glycinin forms homogeneous solutions in water at different concentrations, but undergoes self-coacervation in the presence of sodium chloride[31,32].

Here, we provide an experimental framework to unravel the governing forces behind membrane-condensate interactions at the molecular scale. Hyperspectral imaging and fluorescence lifetime imaging microscopy (FLIM) were combined with the phasor plot analysis[33], whereby 6-acetyl-2-dimethylaminonaphtalene (ACDAN) and 6-dodecanoyl-2-dimethylaminonaphtalene (LAURDAN), two dyes sensitive to water dipolar relaxation are used to probe the local nano-environment. With this approach, we characterize changes in the water dynamics of the protein condensates at increasing salt concentrations, providing information on their interaction with membranes and the associated environmental changes occurring at the molecular level. To resolve the molecular changes occurring at different stages of phase separation, we employed Fourier-transform infrared attenuated total reflection spectroscopy (FTIR-ATR) and Raman microscopy. Our results show that the nano-environment in the protein condensates changes with salt concetration in the medium, leading to different properties in the confined water molecules. These changes affect the local water dynamics, which can be sensed by ACDAN[34–36], providing a means for analyzing the physico-chemical environment in the condensates at nanometer scales. Moreover, we observed that these changes at the nanometer range correlate with changes in the material properties of the condensates measured by bulk rheology.

Wetting transitions are observed when changing the solution salinity, as a consequence of the different affinity between the condensates and the membrane[28]. We investigated the molecular origin of these processes and show that with increasing wetting affinity, the membrane in contact with the condensates becomes dehydrated and more densely packed than the bare condensate-free membrane. Taken together, these results obtained with protein condensates and ATPS systems point to a general mechanism for membrane-condensate interactions, taking a step further in our understanding of the underlying molecular processes.

## Results

### Hyperspectral Imaging of ACDAN senses crowding and water dynamics in the condensate nano-environment

The di-methylaminonaphtalene probes (DAN) are a family of polarity-sensitive dyes, resposive to the solvent dipolar relaxation[37,38] (see Fig. 1a). Because intracellular water is the most abundant dipolar active molecule in biological systems, DAN probes can be employed as sensors for changes in water dynamics in the intracellular space[38–40] and in membranes[38,41]. As shown in Fig. 1b, when the DAN fluorescent probes are excited, there is an increase in the dipole moment of the dye. If the solvent molecules are free to reorient and align with the dye dipole, the emission spectrum of the probe is red-shifted as a consequence of the energy reduction due to solvent relaxation (Fig. 1b, c). Conversely, in an environment where the dynamics of the solvent molecules are restricted, and water cannot reorient around the dye at the excited state, the spectrum is blue-shifted. One of the probes we employ is ACDAN, which is a water-soluble dye that has been previously used to characterize the intracellular environment in yeast[35,39], HeLa cells[40], and most recently for the study of in vivo macromolecular crowding of lenses in zebra fish eyes[34]. Here, using hyperspectral imaging combined with phasor plot analysis[34], we exploit the sensitivity of ACDAN for probing the nano-environment in biomolecular condensates.

Hyperspectral imaging produces a stack of images, in which each pixel contains spectral information, as shown in Fig. 1d. Fourier transform of the spectral data from each pixel allows building a two-dimensional scatter plot with the real and imaginary components (G and S) as axes of the phasor space (see inset in Fig. 1e and Methods). Here, the angular dependence carries information about the center of mass of the emission spectra in all pixels, while the spectra broadening is reflected in the radial dependence of the data. Figure 1e shows the spectral phasor plot of glycinin in water at different concentrations. In water, ACDAN has an emission spectrum centered at 520 nm. Increasing protein-water ratio leads to the spectrum being blue-shifted, reflecting the associated reduction in dipolar relaxation. ACDAN spectroscopic properties provide an accurate quantitative readout of the water activity or confinement, and thus the protein-water ratio and crowding[34]. An easy way of visualizing the spectral shifts consists of extracting the histogram for the pixel distribution along a trajectory defined by the data, i.e. two-cursor analysis (see Methods). Figure 1f shows the histograms of the data at different glycinin concentrations as shown superimposed in Fig. 1e. The x-axis describes the changes in the water dipolar relaxation, which decreases when increasing the protein concentration. For a quantitative and statistical analysis, the center of mass of these histograms, which reflects the degree of dipolar relaxation, can be calculated (see Methods) and plotted as shown in Fig. 1g. The use of phasor plot for analyzing ACDAN fluorescence is important since it avoids biases by assuming a priori a model for data interpretation[34–42]. On the other hand, phasor plot properties such as linear combination and reciprocity principle enable quantitatively addressing the problems under study.

At low salt concentrations (≤43 mM NaCl), glycinin solutions are homogeneous (region R1 in Fig. 2a), but undergo liquid-liquid phase separation at higher salinity (from 43 to 230 mM NaCl, region R2), forming condensates of micrometric size (Fig. 2b), as previously reported[31,32]. Further increasing the salt concentration leads to the dissolution of the condensates and a reentrant homogeneous phase (R3 in Fig. 2a)[31,32]. We tested whether ACDAN hyperspectral imaging and phasor analysis can be used to probe the physicochemical environment of glycinin condensates and solutions at increasing salt concentration. Figure 2c shows the obtained spectral phasor plot, in which the combined data at different salinity fall mainly into two clouds. This binary behavior indicates two very different molecular environments for ACDAN when the protein is phase-separated (region R2) or in a homogeneous phase (R1 and R3). It is important to note that when phase separation takes place, ACDAN preferentially partitions to the

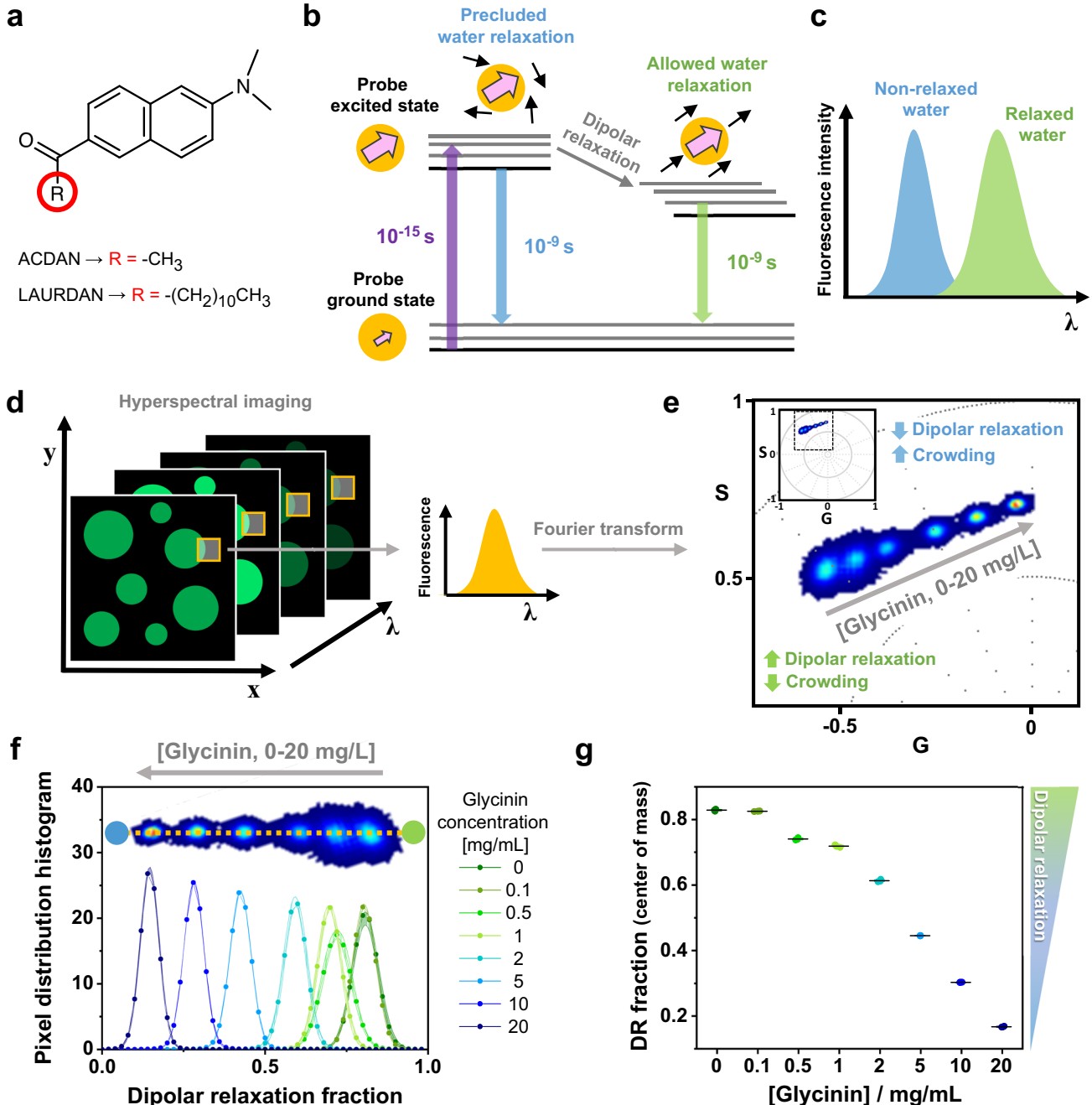

**Fig. 1 | DAN probes solvatochromism allows the quantification of water dipolar relaxation and crowding. a** Molecular structure of the di-methylaminonaphtalene (DAN) probes, highlighting the differences between ACDAN (6-acetyl-2-dimethyla-minonaphthalene, water soluble) and LAURDAN (6-dodecanoyl-2-dimethylamino-naphthalene, partitions in membranes). **b** Perrin-Jablonski diagram illustrating the phenomenon of solvent dipolar relaxation. Upon excitation, there is an increase in the dipole moment of the probe (pink arrow). When the solvent dipoles (black arrows) are free to align with the probe in its excited state, the dipole energy is reduced and the fluorescence emission is red-shifted, as seen in (**c**). **d** Hyperspectral imaging consists of acquiring images at different emission wavelengths to generate a lambda-stack. Each pixel of the final image contains spectral information. **e** Phasor plot of the spectral emission of ACDAN in solutions containing glycinin. The data for different protein concentrations in water (0, 0.1, 0.5, 1, 2, 5, 10 and 20 mg/mL) are super-imposed. Increasing the protein-water ratio results in a blue-shift of the emission, i.e. a decrease in water dipolar relaxation. The inset shows the full spectral phasor plot and the dashed square delineates the fragment that is magnified in the graph. The colors of the pixel clouds indicate the pixel density, increasing from blue to red. **f** Dipolar relaxation fraction histogram from two-cursor analysis for the data shown in (**e**). Data are represented as the mean (dots and lines) ± SD (shadowed contour), $n = 3$ inde-pendent experiments per condition. The inset shows the trajectory displayed by the pixel clouds taken from the phasor plot, from which the histograms were calculated. **g** The center of mass of the histograms for the dipolar relaxation fraction shown in (**f**) is represented as mean ± SD (see Eq. 12), and independent experiments are plotted. Data for panels (**f**) and (**g**) are provided as a Source Data file.

condensate phase, giving almost no signal in the protein-poor phase. The histograms of the dipolar relaxation fraction and the corre-sponding center of mass are displayed in Fig. 2d, e, respectively. The data for both of the homogeneous regimes (R1 and R3) are located at high dipolar relaxation fractions (above 0.5), while the data from

condensates (R2) are consistently low (below 0.5). As shown in Fig. 2d, e, there is a pronounced decrease of dipolar relaxation occurring with the onset of phase separation, reaching a minimum value at 75 mM NaCl and then reversing towards higher dipolar relaxation values. These results show not only that ACDAN senses increased crowding

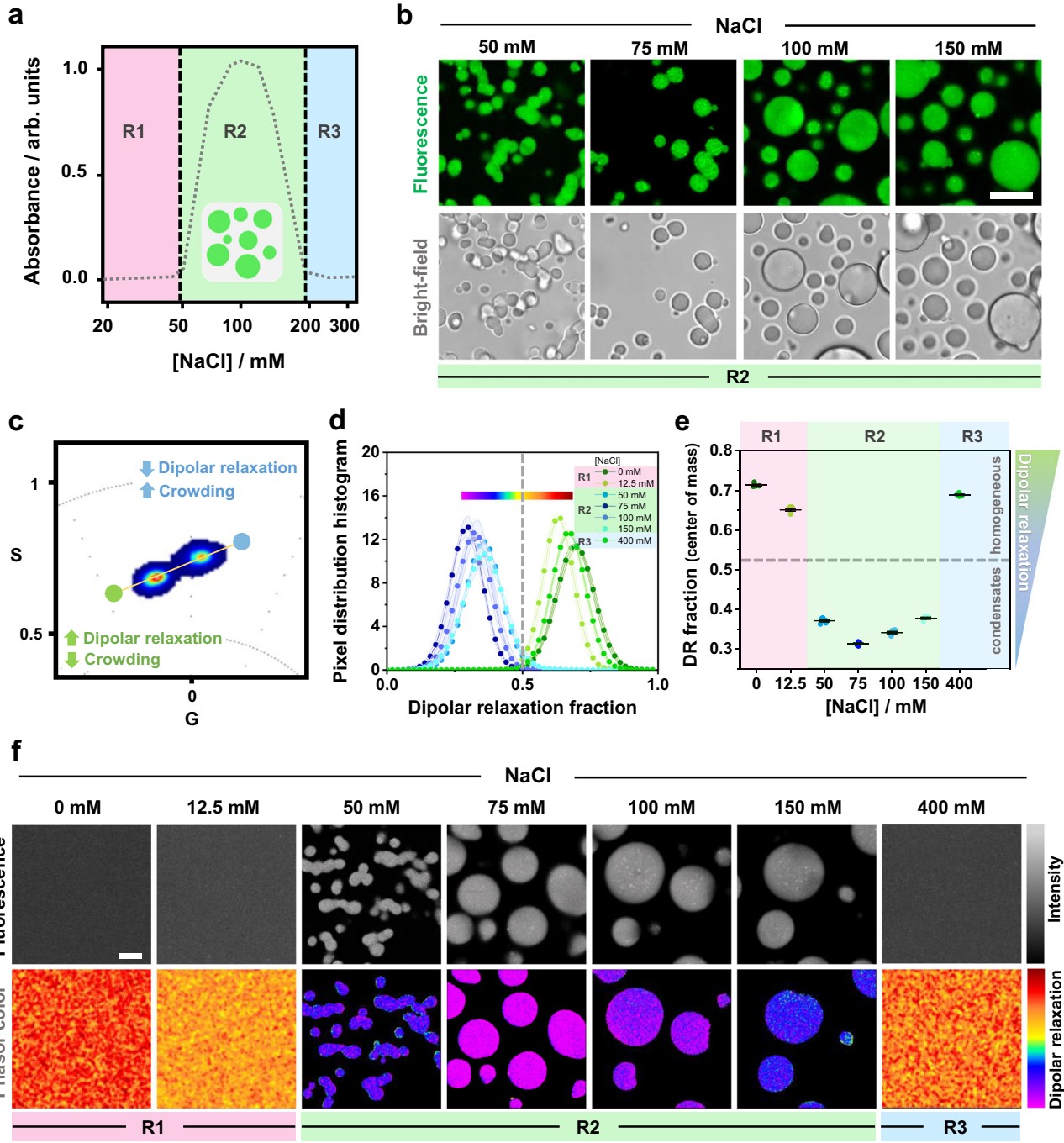

**Fig. 2 | Characterization of glycinin phase separation using ACDAN fluorescence spectral phasor analysis. a** Glycinin exhibits phase separation at intermediate NaCl concentrations as shown from absorbance measurements (*x*-axis is in log scale). In regions R1 and R3 glycinin solutions are homogeneous, while in R2, phase separation occurs (adapted from Chen et al. [31]). **b** Confocal cross sections and brightfield images of glycinin condensates at different NaCl concentrations (within region R2 in (**a**)). Scale bar: 10 µm. **c** ACDAN spectral phasor plot for glycinin in water and in solutions at 12.5, 50, 75, 100, 150, and 400 mM NaCl. The data for the different conditions fall into two main pixel clouds. **d** Histograms obtained from two-cursor analysis showing the distribution of pixels along the yellow line depicted in (**c**). Starting from the blue circular cursor and moving towards the green circular cursor, crowding decreases and dipolar relaxation increases. The distance between the cursors is used as the *x*-axis showing the dipolar relaxation fraction. An arbitrary line was placed at *x* = 0.5 to separate the two groups of data corresponding to homogeneous (R1, R3 with x roughly above 0.5) and condensate (R2

with x roughly below 0.5) regions in the phase diagram. Each histogram is shown as mean ± SD, *n* = 5 independent experiments per condition. The continuous color map used to visualize the degree of dipolar relaxation in the images in (**f**) is shown above the histograms. **e** Center of mass of the dipolar relaxation (DR) distributions shown in (**d**) for glycinin in water and at the indicated NaCl concentrations. When the system is homogeneous, data falls above 0.5 (horizontal dashed line corresponding to *x* = 0.5 in (**d**)), indicating stronger dipolar relaxation. When the system is phase-separated, the data point to weaker dipolar relaxation (crowded environment). Zoomed plots of d and e can be found in Supplementary Fig. 1. Independent experiments are plotted as circles, the lines indicate mean ± SD. **f** Examples of ACDAN fluorescence intensity images (upper panels) and corresponding images visualized with a continuous color map defined in (**d**) indicating dipolar relaxation. Scale bar: 5 µm. Data for panel (**e**) are provided as a Source Data file.

and constrained water relaxation in condensates (R2) compared to the homogeneous regimes (R1 and R3), but it can also be sensitive to changes within the condensate regime under different conditions (see zoomed plots in Supplementary Fig 1).

An additional feature of the phasor approach is the reciprocity principle[33,42], by which a continuous color scheme (see color bar in Fig. 2d) can be used to visualize the images according to the phasors distribution (Fig. 2f), providing a qualitative spatial view of the changes taking place. Here, the same trend obtained in the quantitative analysis is followed: dipolar relaxation decreases with increasing salinity, reaching a minimum (more purple colored pixels) at 75 mM NaCl and then reversing to a more polar environment (red colored pixels). Altogether, quantification of ACDAN dipolar relaxation shows that this probe, as well as phasor analysis, are sensitive to liquid-liquid phase separation and show different nano-environments within the different conditions, even within the homogeneous regime.

### Protein secondary structure rearrangements modify the water dynamics in condensates

Glycinin is a hexamer with two symmetric trimers stacked together and a total molecular weight of 360 kDa[43]. Each monomer has an acidic and a basic polypeptide linked by a single disulfide bridge[43]. The salinity-triggered phase separation in glycinin solutions suggests molecular rearrangements between the basic and acidic residues[31]. However, this mechanism is very general, offering only a rough interpretation of the condensation phenomenon, while the structural changes in the protein between and within the different phase regimes shown in Fig. 2a remain unexplored. In order to understand how the changes sensed by ACDAN are related to protein structural rearrangements, we used label-free spectroscopic techniques, namely FTIR-ATR and Raman microscopy. These tools provide information on the protein secondary structure and the environment within the condensates. Figure 3a shows examples of the protein Amide I band obtained by FTIR-ATR for the different regions (R1, R2 and R3, see Fig. 2). When glycinin phase-separates at intermediate salt concentrations (R2), the band shifts towards higher wavenumbers (see inset in Fig. 3a), and is downshifted towards lower wavenumbers at higher salt concentrations for the reentrant homogeneous regime (R3). These shifts imply structural changes in the protein that become more evident when taking the second derivative of the spectra (see Supplementary Fig. 3a). Figure 3b summarizes the secondary structure percentage content of glycinin at the different NaCl concentrations (see methods and Supplementary Fig. 2 and 3 for details). The differences between conditions can be more clearly appreciated by inspecting the relative change in secondary structure content compared to that of glycinin in the absence of NaCl, as shown in Fig. 3c. Upon phase separation, the percentage of random coil motifs increases and the alpha-helix content decreases. Meanwhile the beta-sheet content changes are less pronounced and do not show a clear trend (Supplementary Fig. 3b). The increase in random coils with the onset of phase separation is expected, since this process is usually driven by the interaction of intrinsically disordered regions within proteins[44–46]. At high salt concentration, the secondary structure motifs of glycinin seem to rearrange back to the structure observed in the absence of NaCl. This result supports the ACDAN data (compare with Fig. 2d, e and Supplementary Fig. 1), which showed different spectral behavior at different salt concentrations even within the homogeneous regimes (R1, R3). Widely used in material and food sciences, glycinin solutions in water have been extensively studied using FTIR-ATR[47]. Here, we report the secondary structural changes of glycinin in various conditions, under which phase separation occurs.

Considering that ACDAN is sensitive to water dipolar relaxation and that the protein secondary structure is changing at different salt concentrations, it is likely that the properties of the confined water

inside the condensates are altered in turn[34,35,48]. To confirm this, we employed Raman spectroscopy, which allows measuring changes in the collective structure of water by analyzing the band corresponding to water OH vibrations engaged in hydrogen bonding[48,49]. While in FTIR spectroscopy the water band interferes with the protein signal, in Raman the signal for the water OH vibrations appears at higher wavenumbers, away from the protein fingerprint region. Furthermore, when combined with microscopy, Raman spectroscopy provides images, in which each pixel contains spectral information. Once obtained, it is possible to visualize the image according to the intensity of a given spectral band. Figure 3d shows the Raman image of a condensate segment and the continuous phase surrounding it colored according to the band intensities of Amide I ($1600–1700\ cm^{-1}$) and water ($3000–3700\ cm^{-1}$). The water band intensity decreases towards the interior of the condensate while the protein one increases in the same direction (see color bar and intensity profiles in Fig. 3d). This result is understandable as protein/water ratio is higher in the condensates compared to the continuous phase. When analyzing the water band spectra for glycinin the different salinities, we noticed the intensity of the water band changed, as well as the contribution of the bands corresponding to different hydrogen bonding states[49,50] (see Fig. 3e). The two main contributions in the water band correspond to tetra-coordinated and di-coordinated water molecules and are located at $3225\ cm^{-1}$ and $3432\ cm^{-1}$, respectively[49,50]. In liquid water, the $3432\ cm^{-1}$ band dominates the signal. The $3225\ cm^{-1}$ becomes dominant, however, when, for example, water freezes, causing a spectral shift towards lower wavenumbers due to the increased strength of the tetra-coordinated hydrogen bonding[51]. The band intensity changes, together with the Raman spectral shift can be quantified by using the generalized polarization function (GP, see Methods), which provides information on the water collective behavior[48]. Figure 3f shows that the GP for the Raman water band increases with salt concentrarion, presenting a maximum at 100 mM NaCl and decreasing at higher salinity. Altogether, these results indicate that the changes in the protein secondary structure modulate the degree of water hydrogen bonding, and that ACDAN is sensitive to these nano-environmental changes occuring in its immediate surroundings (compare the trends in Figs. 2e and 3e), providing a fingerprint of the condensate under different conditions.

### Glycinin condensates mechanical and rheological properties are tuned by salinity

Having proved that changing the salt concentration leads to rearrangements of the protein secondary structure and modifies the water nano-environment within condensates, we tested whether these changes are reflected in the mechanical and rheological properties of the condensates. As indicated in the previous section, glycinin is a hexamer of high molecular weight (360 kDa). As such, it forms highly viscous condensates[28,31] with a viscosity on the order of $10^3\ Pa.s.$ similar to that of the nucleolus[52,53]. Determining the diffusion coefficient of glycinin in the condensates using techniques like Fluorescence Recovery After Photobleaching (FRAP) is hindered by their high viscosity, as shown in Supplementary Fig. 5a. While the recovery curves exhibit reproducibility and discernable trend, these data do not allow quantifying the protein mobility within the condensates. Another conventional approach to characterize the material properties of condensates consists in monitoring their coalescence over time[54]. Glycinin condensates display coalescence within minutes[31] (see Fig. 4a), with a relaxation time depending on the salinity, as shown in Fig. 4b. The slopes of the curves in Fig. 4b yield the inverse capillary velocity ($\eta/\gamma$), relating the viscosity ($\eta$) and surface tension ($\gamma$) of the condensates. As evidenced in Fig. 4c, the inverse capillary velocity changes for condensates at different salt concentrations, clearly indicating that the condensate material properties are dependent on the salinity.

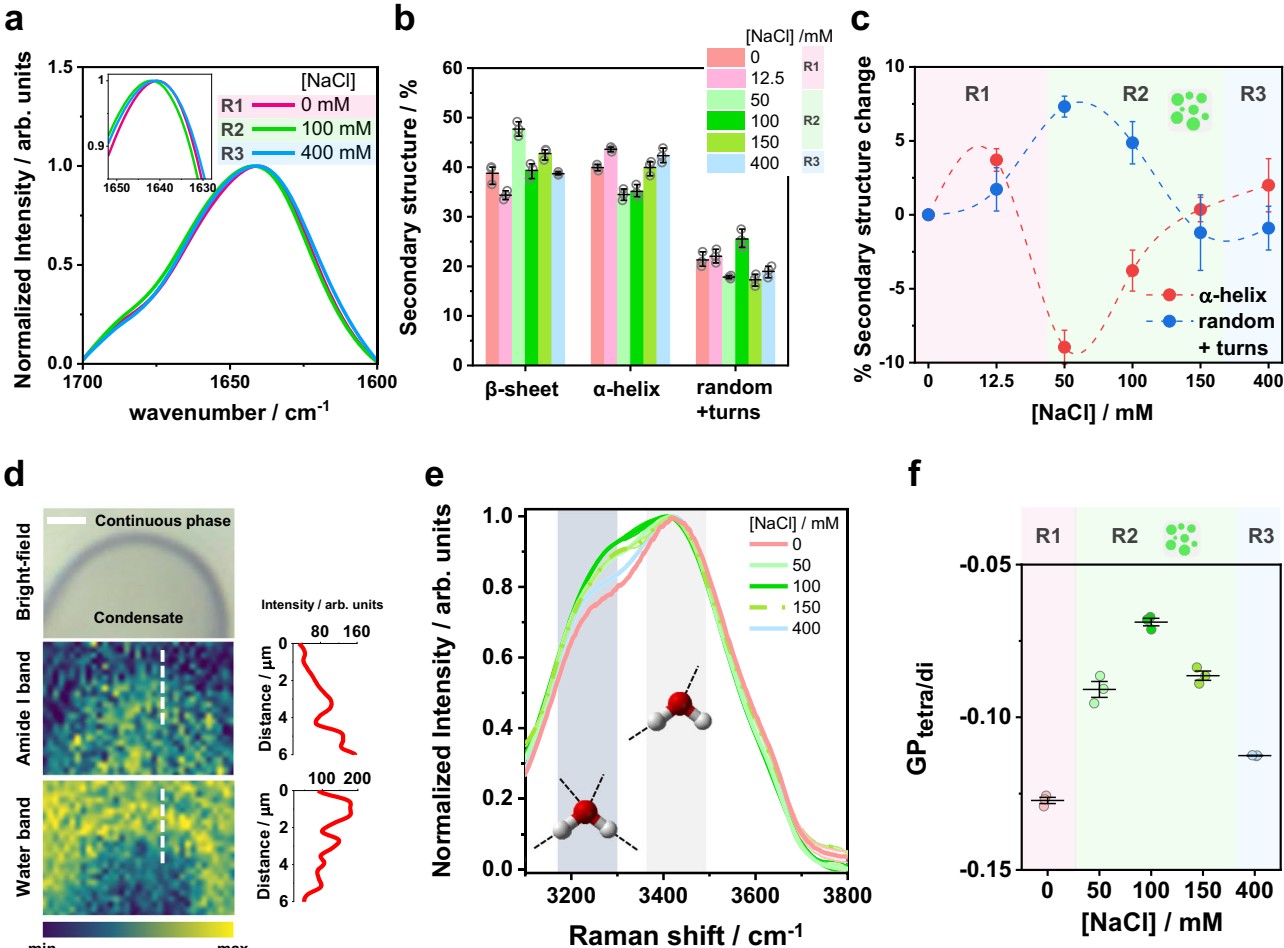

**Fig. 3 | Glycinin secondary structure changes with salt concentration and modifies the water environment inside condensates. a** Examples of FTIR-ATR spectra of the Amide I band of glycinin in different regions of the phase diagram in Fig. 2a: R1 (0 mM NaCl), R2 (100 mM NaCl), R3 (400 mM NaCl); see Supplementary Figs. 2 and 3 for details. The inset shows a zoomed region highlighting the spectral shifts. **b** Secondary structure content for glycinin at different conditions obtained by ATR-FTIR analysis. Individual data points are shown (circles) together with the mean ± SD values (black lines). $n = 3$ independent experiments. **c** Percentage change in secondary structure motifs for the different salinity conditions relative to the structure of glycinin in salt-free water. The plotted data were obtained by subtracting the average values for each condition shown in (**b**). The error bars were calculated as $\sigma_{i-j} = \sqrt{\sigma_i^2 + \sigma_j^2}$, where $\sigma$ is the standard deviation. Major secondary structure rearrangements of the protein while changing salinity are associated with the α-helix and random+turns content. **d** Raman microscopy image of a section of a single condensate at 100 mM NaCl. Pixel color is mapped to the intensity of the Amide I band (middle image) or water band (bottom image) as indicated by the color bar. Intensity profiles shown next to the images were acquired along the white dashed lines in the images. Scale bar is 3 μm. **e** Raman spectra of the water band at different NaCl concentrations. Lines are mean values and SD is shadowed ($n = 3$ independent experiments). The regions in gray indicate the main bands around 3225 and 3432 cm⁻¹ corresponding to tetra-coordinated and tri-coordinated water molecules respectively as shown by the cartoons. **f** Spectral changes in the Raman water band quantified with the $GP_{tetra/di}$ function, calculated as indicated in Eq. (13). See Supplementary Fig. 4 for further details. Independent experiments are plotted as circles and the lines represent mean ± SD. The background in panels (**c**) and (**f**) was colored to represent the different regions of the phase diagram in Fig. 2a. Data for panels (**a**–**c**) and (**e**, **f**) are provided as a Source Data file.

To further characterize the material properties of condensates, we performed bulk rheology measurements of the protein-rich phase at different salt conditions (see Methods). Figure 4d shows the change in the phase angle ($\delta$) when varying frequency. The phase angle, defined as the tangent of the ratio between the loss and storage moduli, is a relative measure of the contributions of a material's viscous and elastic characteristics to its overall mechanical properties[55]: Purely elastic solids exhibit $\delta=0°$ while purely viscous liquids have $\delta=90°$. A plot of the phase angle at different frequencies provides an indication of the type of material under study and how it responds to different mechanical stresses (see Supplementary Fig. 6 for data interpretation). At all salt concentrations the condensate phase behaves as a viscoelastic liquid. Figure 4e−g shows the storage ($G'$) and loss ($G''$) moduli of the condensates, which together make up the complex shear modulus, $G^* = G' + iG''$, measured as functions of the oscillatory shear frequency for the salt concentrations under study.

Quantities such as the complex viscosity ($\eta^*$) and the terminal relaxation time ($\tau_m$) can be calculated from the complex modulus (see Supplementary Fig. 5b, c). The loss modulus, $G''$, describing the viscous behavior of a material, will dominate at all frequencies for a purely viscous condensate, while the storage modulus, $G'$, will dominate for an elastic condensate[54]. In Fig. 4e−g, $G''$ dominates at short frequencies and long timescales, meaning the condensates behave more like liquids, while $G'$ dominates at high frequencies and short timescales, with the condensates exhibiting more solid-like behavior. This general behavior is consistent across all tested salt concentrations. From the linear part of the loss modulus in the low-frequency range, one can obtain the condensate zero-shear viscosity ($G'' = \omega\eta$, where $\omega$ is the frequency). Figure 4h shows that the obtained viscosities change considerably with salinity spanning almost two orders of magnitude. The value measured at 100 mM NaCl is on the order of kPa.s, which is consistent with data obtained on individual glycinin droplets using

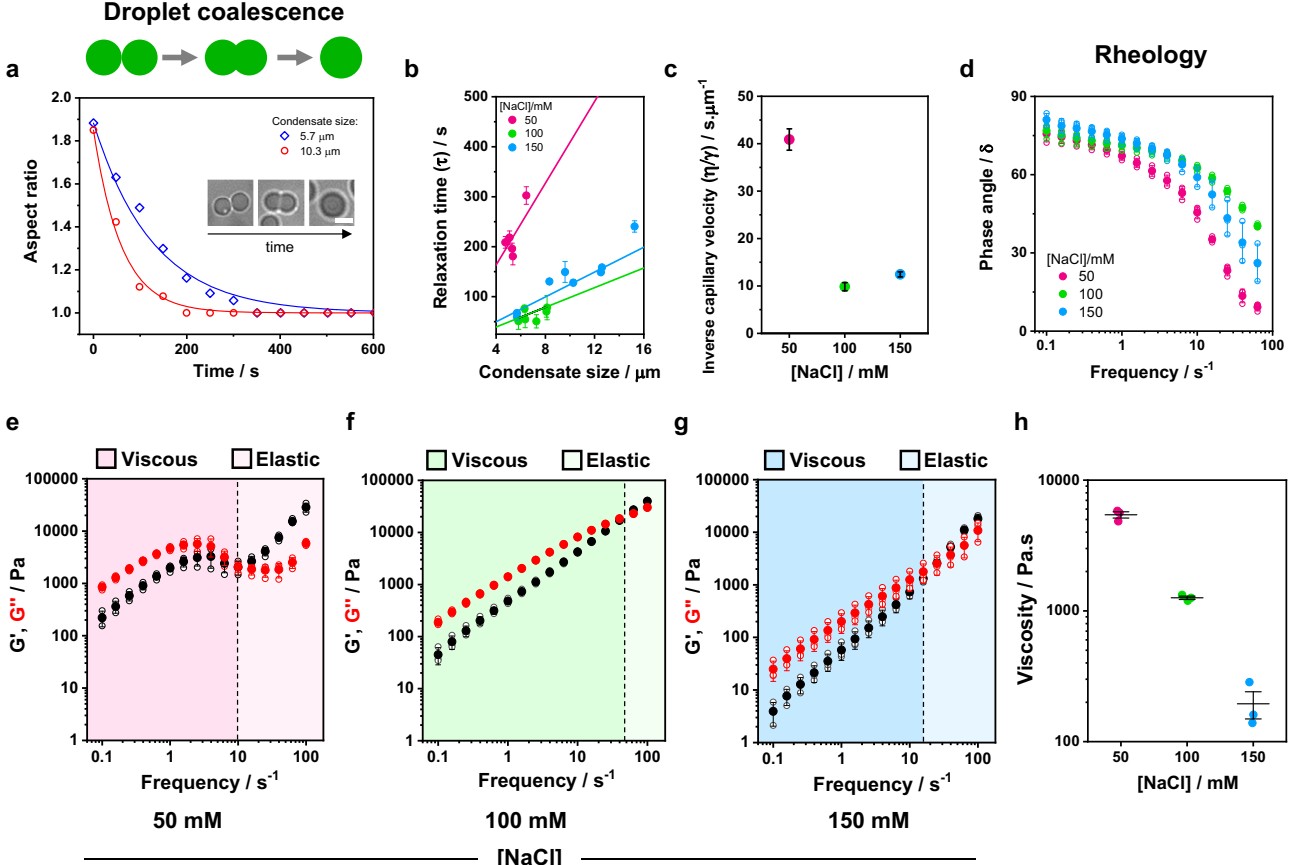

**Fig. 4 | Glycinin condensates material properties change with salinity.** The material properties of condensates at 50 mM (pink), 100 mM (green) and 150 mM (blue) NaCl were evaluated with different approaches. **a** Examples of aspect ratio vs time for coalescing condensates of different sizes. Glycinin condensates coalesce within minutes, displaying different characteristic relaxation times according to size and NaCl concentration. The data are fitted with the function $y = 1 + (y_0 - 1) \cdot \exp(-x/\tau)$, where $\tau$ is the characteristic relaxation time. The inset shows an example of condensate coalescence at 100 mM NaCl. The scale bar is 5 µm. **b** Plot of the relaxation time vs. the final condensate diameter. Values are shown as mean ± SD, $n = 19$ independent experiments. Solid lines are fits to the linear equation: $y = \frac{\eta}{\gamma}x$, where the slope, $\frac{\eta}{\gamma}$, is the inverse capillary velocity. **c** Inverse capillary velocity obtained from (**b**) for varying salt concentrations indicate that the material properties of the condensates are modulated by salinity conditions. The data plotted correspond to mean ± SD values of the slopes of the curves shown in

(**b**). **d**–**h** Rheology measurements of glycinin condensates at different salt concentrations display changes in the material properties. **d.** Phase angle vs frequency for glycinin condensates at the different conditions. In all cases condensates behave as viscoelastic liquids (see Supplementary Fig. 6). Independent experiments are plotted as open circles and the mean ± SD are shown in solid circles ($n = 3$). **e**–**g** Plots showing the average storage and loss modulus ($G'$, black, and $G''$, red) vs frequency for glycinin condensates at the indicated salinities. Independent measurements are plotted as hollow circles and the mean ± SD are shown in full circles ($n = 3$). Shaded regions represent the the dominant viscous or elastic regime, as indicated. The crossover frequency (black dashed line) is equal to $1/\tau_m$, where $\tau_m$ is the terminal relaxation time. **h** Zero-shear viscosity for the condensate phase at different salt concentrations. Independent measurements are shown as circles, and lines represent mean values ± SD. Data for panels (**a**–**h**) are provided as a Source Data file.

microscopy approaches[28,31]. The frequency-dependent mechanical response displayed by glycinin condensates is similar to that of typical Maxwell fluids, presenting a single crossover point between the viscous and elastic regimes for the frequencies tested[56,57]. This crossover point occurs at different frequencies, depending on salt concentration, further indicating that the material properties of the condensates are changing with salinity. The terminal relaxation time, $\tau_m$, can be calculated as the inverse of the crossover frequency, and indicates the average reconfiguration time of the protein network within the condensate. Supplementary Fig. 5c shows that the value of $\tau_m$ follows the order $\tau_m(50 \text{ mM}) > \tau_m(150 \text{ mM}) > \tau_m(100 \text{ mM})$ NaCl. The obtained values are similar to those found for Arginine/Glycine-rich (R/G) peptides containing the RGRGG motif[56]. Interestingly, this trend is similar to that observed for the inverse capillarity in Fig. 4c, and can be related also to the degree of water hydrogen bonding shown in Fig. 3e. Altogether, these results indicate that the salt-driven protein structural rearrangement that leads to changes in the water hydrogen bonding within the condensates results in altered mechanical and rheological properties of the condensates.

## Dehydration and lipid packing increase upon membrane wetting by biomolecular condensates

Protein condensates can wet membranes, promoting their remodeling[27,28], as also shown for polypeptide-based coacervates[58,59] and PEG/dextran condensates[25,60]. However, a molecular view of this interaction is still missing. We have previously shown that, in R2 regime of the glycinin phase diagram, condensate wetting affinity to membranes is enhanced at higher NaCl concentration[28]. Here we use LAURDAN, the DAN probe including a lipid (lauroyl) chain (see Figs. 1a, 5a), to investigate the effect of wetting on molecular rearrangements in the membrane. LAURDAN is a fluorescent probe sensitive to membrane polarity and water dipolar relaxation[38,41]. While the membrane polarity is related to the apparent dielectric constant at the LAURDAN location in the membrane, the dipolar relaxation corresponds to the re-orientation of the water molecules around the dye in response to the increase in LAURDAN's dipole moment upon excitation (see Fig. 1b). The photophysics behind LAURDAN fluorescence in lipid membranes has been extensively described and reviewed for over 30 years[38,41,61,62]. In membranes, LAURDAN fluorescence is responsive to

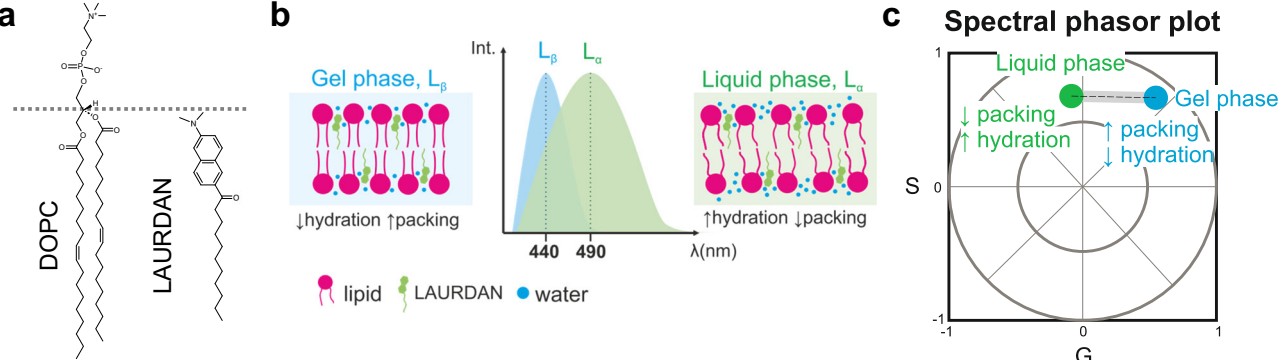

**Fig. 5 | LAURDAN phasor analysis reports on membrane packing and hydration. a** Molecular structures of DOPC and LAURDAN. The dashed line indicates the approximate relative locations of the lipid and the dye from the bilayer center (-1.5 nm[104]). **b** Scheme illustrating the spectral shifts for LAURDAN in membranes with different properties: highly packed and dehydrated membranes, like those in the liquid-ordered ($L_o$) or in the gel-phase state ($L_\beta$) will present a blue-shifted spectrum with a maximum located near 440 nm. Membranes in a liquid phase state ($L_\alpha$) will present a red shifted spectrum with a maximum centered around 490 nm[38,41]. **c** Sketch of a spectral phasor plot showing the trajectory for LAURDAN fluorescence in membranes. Spectra corresponding to different degrees of water penetration will fall within the linear trajectory between the two extremes for the liquid and the gel phases[33,38,66]. Any deviation from this trajectory would indicate the presence of a third component, as shown in Supplementary Fig. 7b.

the dynamics of a few water molecules in the immediate environment of the bilayer, nearby the glycerol backbone of the glycerophospholipids[41], as illustrated in Fig. 5. For this reason, LAURDAN has been widely used to assess the membrane phase state and hydration level[38,41,48,63,64].

Lipid bilayers in the liquid phase ($L_\alpha$) are less packed, more hydrated and with higher polarity compared to those in a gel ($L_\beta$) or liquid ordered ($L_o$) phase. They present greater dipolar relaxation, since the water molecules are able to reorient around the LAURDAN moiety during its excited state (see Fig. 1b). Membranes in the $L_\beta$ or $L_o$ phases are highly packed and dehydrated with low polarity, and the few water molecules present in the bilayer are not able to reorient while the dye is in the excited state. This is summarized in Fig. 5b, showing that LAURDAN fluorescence displays a big spectral shift (-50 nm) between the liquid phase and the gel phase. Between these extremes, there is a broad range of intermediate membrane hydration states that LAURDAN is sensitive to, and are related to different degrees of lipid packing. LAURDAN has been shown to be sensitive to small changes in membrane packing, even between lipids in the same phase state[48,63,65,66]. Here, we use the term "fluidity" referring to the order of the membrane headgroup-chain interface for a phospholipid bilayer. We consider that any process that affects the rotational or translational movement of lipids is changing the fluidity. The direct relationship between the LAURDAN spectral properties and the order of the glycerol backbone interface was recently confirmed with nuclear magnetic resonance spectroscopy (H-NMR)[64].

The spectral phasor plot for the analysis of LAURDAN fluorescence has proven to be an outstanding and straightforward tool for the interpretation of the phenomena taking place at the membrane interface[38,65]. Due to the linear combination properties of the Fourier space, LAURDAN fluorescence in membranes produces a linear trajectory in the spectral phasor plot (see an example in Supplementary Fig. 7a), reflecting different packing and hydration states. The trajectory extremes correspond to the liquid and the gel phases[38,48,63,65,66], as illustrated in Fig. 5c. Deviations from this trajectory would indicate the presence of a third component, as exemplified in Supplementary Fig. 7b. Note that such effects cannot be distinguished when using the classical approach of LAURDAN general polarization, since it assumes a priori a two-state model[67].

We used LAURDAN and spectral phasor analysis to address the effect of condensate wetting on membrane packing and hydration. Figure 6a shows a scheme of the different partial wetting morphologies that can be obtained when condensates interact with GUVs[28]. The

wetting is characterized by the change in the contact angles[23,28] (Fig. 6a and Supplementary Fig. 8); note that the optically measured contact angles only apparently reflect the wetting phenomenon[62], but can be used to deduce the geometric factor characterizing the affinity of the condensate for the membrane[28] (see Supplementary Fig. 8). The bottom panel in Fig. 6a shows examples of DOPC GUVs in contact with condensates at different NaCl concentrations. With increasing salt concentration, the condensates spread more on the vesicles, meaning that the wetting affinity is stronger[28]. To evaluate the effect of condensate wetting on membrane properties, we prepared DOPC GUVs containing 0.5 mol% LAURDAN, put them in contact with unlabeled glycinin condensates, and applied the spectral phasor analysis to these membrane-condensate systems. Using the two-cursor analysis, we could separate the contributions of the membrane in contact with the condensate from the condensate-free (bare) membrane at a single vesicle level, as shown in Fig. 6b. Remarkably, the membrane in contact with the condensate displays increased lipid packing (indicated as lower fluidity fraction in Fig. 6b) compared to the bare membrane. It is important to highlight that the measured fluorescence is only from LAURDAN present in the membrane segments, as shown in Supplementary Fig. 9, and the contribution from protein autofluorescence is negligible.

Next, we evaluated how this effect changes when the interaction between the membrane and the condensate is stronger, i.e. when the membrane wetting affinity increased. Figure 6c shows the fluidity fraction histograms and the corresponding center of mass for the membrane segments in contact with condensates with increasing salt concentration, i.e. higher wetting affinity. The analysis shows that increasing the wetting strength results in higher lipid packing and membrane dehydration. In contrast, the condensate-free membrane segment remains unaltered as evidenced by the negligible change in the fluidity fraction (Fig. 6d).

The strongest interaction between glycinin condensates and the membranes as determined by the geometric factor[28] is observed at 180 mM NaCl (complete wetting, Supplementary Fig. 8b); however, the difference in the fluidity fraction between this salt condition and 150 mM NaCl is negligible (Supplementary Figs. 8c, d).

Glycinin phase separation can be triggered not only by salinity variations but also by changes in pH or temperature changes[31]. In addition, glycinin condensates can transform into hollow ones via a quick change of the salinity of the milieu[31] (Supplementary Fig. 10a). We have previously shown that hollow condensates can wet and reshape the membrane in a similar manner than isotropic,

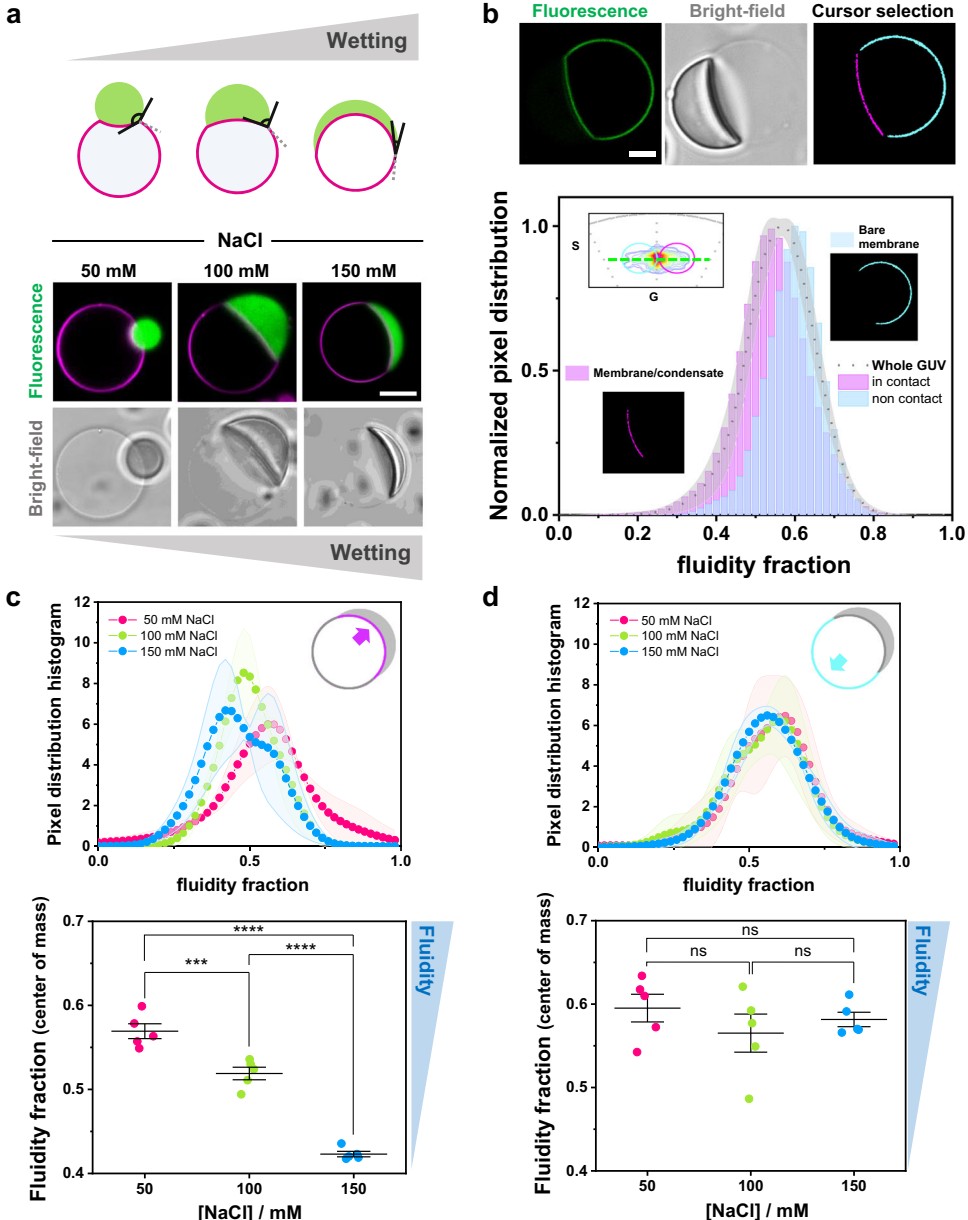

**Fig. 6 | Membrane wetting by condensates increases lipid packing and membrane dehydration. a** Biomolecular condensates can wet membranes[23,28], and the increase in wetting can be observed as an increase in the spreading of the droplet over the membrane, as well as a decrease in the contact angle as illustrated in the sketch (upper panel). Examples of membrane wetting (DOPC GUVs labeled with 0.1 mol% Atto 647N-DOPE, magenta) by glycinin condensates (FITC labeled, green) at the indicated NaCl concentrations (bottom panel). Scale bar: 10 μm. **b–d** DOPC GUVs labeled with LAURDAN (0.5 mol%) in contact with unlabeled condensates. Using LAURDAN fluorescence spectral phasor analysis, we can segment the vesicle and separate the contributions of the membrane in contact with the condensate and the bare membrane on the same GUV. **b** Example analysis for a single DOPC GUV labeled with 0.5% LAURDAN (green) in contact with an unlabeled condensate (at 100 mM NaCl). The cursor-colored segments and the corresponding histograms are shown in the bottom panel. The part of the membrane in contact with the condensate is more packed (with lower fluidity fraction) than the bare membrane. Distributions were normalized for clarity. Scale bar: 10 μm. **c, d** Histograms of the pixel distribution (top panel), and center of mass of the distributions (bottom panel) for the membrane segment in contact with the condensate (**c**) and for the bare membrane segment (**d**), at the indicated NaCl concentrations. The sketches indicate the part of the membrane being analyzed. Data are shown as mean ± SD ($n = 5$ GUVs per condition). The statistical analysis was performed with One-way ANOVA and Tukey post-test analysis ($p < 0.0001$, **** | $p < 0.001$, *** | $p < 0.01$, ** | $p < 0.05$, * | ns = non-significant). Data for panels (**b–d**) are provided as a Source Data file.

homogeneous condensates[31]. Here, we tested whether hollow condensates could induce comparable membrane packing to isotropic condensates (Supplementary Fig. 10c, d). We observed that the membrane segment wetted by the hollow condensates presents an increased packing when compared to the bare membrane, as shown in Supplementary Fig. 10c. We then compared the effect on the membrane that hollow and isotropic condensates have at the same NaCl concentration and observed that hollow condensates can produce a higher increase in the lipid packing (Supplementary Fig. 10d). This difference suggests that hollow condensates might present different protein organization or composition[68] compared to isotropic ones, leading to different material properties. This was confirmed when evaluating ACDAN spectral response in hollow condensates. We observed that they show reduced dipolar relaxation compared to isotropic condensates at the same salinity condition (Supplementary Fig. 10e), indicating a distinct nano-environment.

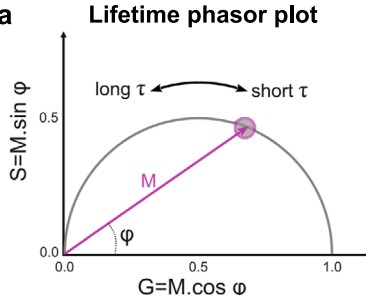

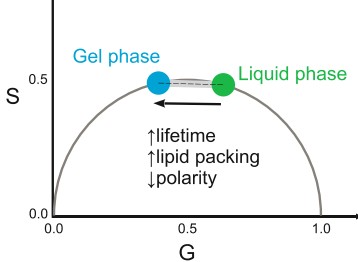

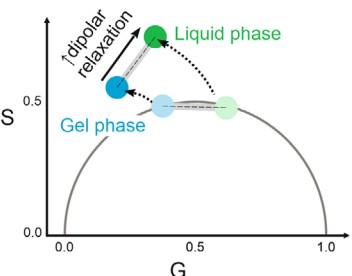

**Fig. 7 | Lifetime phasor plots allow discriminating between polarity and dipolar relaxation changes for LAURDAN decay in membranes. a** Lifetime phasor plot. The modulation (M) indicates the distance of the phasor point from the origin (0:0), and the phase angle ($\varphi$) the decrease or increase in lifetime ($\tau$). Together, these parameters determine the position of the phasor point in the universal semicircle. **b** When excited-state processes take place, M remains the same, but due to the delay in the emission ($\Delta\varphi$), the phasor points appear outside the universal circle as the plot rotates[33,42,69,71]. **c, d.** Using different bandpass filters, the contributions of the polarity and the dipolar relaxation can be split for LAURDAN decay. **c** The lifetimes measured through the blue channel (416–470 nm) give information about the change in lipid packing. The sketch illustrates how the lifetime changes between liquid and gel phases. Linear combination rules apply and the changes can be quantified in the same manner as described for spectral phasors (Fig. 1). All intermediate packing states will fall within the linear trajectory between these two extremes. **d** The green channel (500–600 nm) for LAURDAN fluorescence provides information on water dipolar relaxation processes, and the phasor points fall outside the universal semicircle. For membranes in the liquid phase state, dipolar relaxations are more pronounced because the water molecules have enough time to reorient around the LAURDAN moiety (see Fig. 1b), while for gel and liquid ordered phases dipolar relaxation is less pronounced.

## Membrane hydration and packing changes are hallmarks of membrane-condensate interactions

To test whether the observed changes are independent of the condensate system, we evaluated the effect of wetting on membrane packing in the well-studied aqueous two-phase system (ATPS) consisting of a mixture of PEG and dextran[23,25]. Here, we used LAURDAN fluorescence with FLIM and phasor analysis to unravel the changes in lipid packing and hydration. This methodology presents an advantage over the spectral phasor approach since, given the photophysics of LAURDAN, we can independently study the polarity and the water dipolar relaxation changes by using two channels (see Methods) for collecting the emission signal[38,48,66,69,70], as explained below. In addition, the acquisition mode of FLIM allows the accumulation of several frames, providing a reasonable resolution for ATPS systems, since they presented a low signal-to-noise ratio.

FLIM measurements involve exciting the sample with a pulsed laser, and recording the emission intensity vs time at each pixel. The excited fluorophores will give rise to a modulated emission shifted in phase relative to the exciting light[33]. The resulting lifetime information can be represented in a phasor plot, as shown in Fig. 7a. The semicircle is called the universal semicircle, and all mono-exponential lifetimes will fall within it[33]. The modulation (M) represents the distance of the phasor point from the origin, while the phase angle ($\varphi$) determines the position of the phasor point in the graph (Fig. 7a, Eqs. 8–11). The lifetimes ($\tau$) increase counter-clockwise; when $\varphi$ increases and M decreases, the lifetime is longer. Figure 7b shows that processes occurring during the excited state of the dye, such as dipolar relaxations, Förster resonance energy transfer (FRET) or excimer formation, can be easily identified because the phasor data will appear outside the universal semicircle[33,42,71]. This results from a delay in the emission due to the time required for these processes to take place. In the case of dipolar relaxations, the delay is caused by the time required to reorganize water molecules around the LAURDAN excited dipole (see Fig. 1b)[33,42,71]. The additional phase shift in the excited state of the probe, results in an overall rotation of the plot moving the phasor distribution outside the universal semicircle[33,69] (Fig. 7b). Using a "blue" filter (i.e. collecting the emission at 416–470 nm) allows measuring changes in polarity due to changes in the apparent dielectric constant (Fig. 7c), while with a "green" filter (500–600 nm) we can isolate the dipolar relaxation contributions to the lifetime[33,38,42,69] (Fig. 7d, Supplementary Fig. 11).

All properties of the Fourier space described for the spectral phasors, such as the linear combinations and the reciprocity principle, also apply here, allowing the quantification of the observed changes (see Fig. 1).

In this manner, we used the phasor analysis for FLIM to quantify changes in packing and dipolar relaxation in vesicles in contact with ATPS. Deflation of vesicles encapsulating homogeneous solutions of PEG and dextran can result in liquid-liquid phase separation in the vesicle interior. This is achieved by increasing the osmolarity of the external milieu. Water permeates out of the vesicle due to the osmotic gradient, and in consequence, the polymers inside become concentrated, driving the phase separation[60,72]. We used a microfluidic setup[72] that allows us to work at single-vesicle level and exchange the buffer conditions in a stepwise and controlled manner. In Fig. 8a, the degree of deflation, r, indicates the ratio between the osmolarity outside and the initial osmolarity inside the vesicles. A small degree of deflation produces excess membrane area, which is stored in tubes protruding towards the vesicle interior. These tubes are stabilized by spontaneous curvature generation from asymmetric PEG adsorption[73]. However, at higher deflation ratios ($r \geq 1.3$), phase separation occurs resulting in bud formation and the accumulation of nanotubes at the

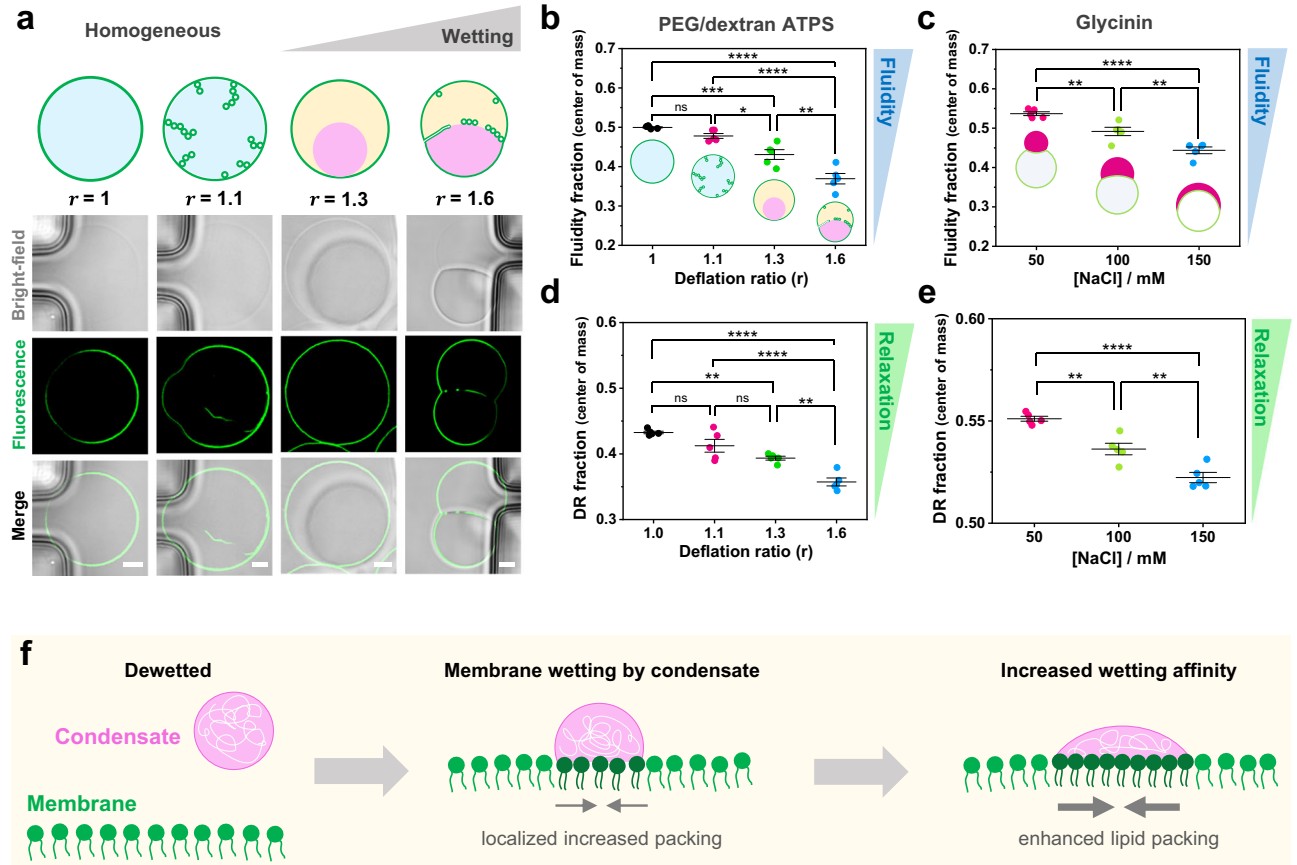

**Fig. 8 | Tuning membrane lipid packing and hydration is a general mechanism underlying wetting by condensate droplets. a.** DOPC vesicles labeled with 0.5 mol% LAURDAN filled with a PEG/dextran solution (ATPS) undergo morphological transformations due to the increase in membrane wetting affinity by the polymer-rich phases. Vesicle deflation with associated increases in the internal polymer concentration result from exposure to solutions of higher osmolarity. This leads to tube formation and subsequent phase separation in the vesicle (see Methods for further details). The dextran-rich phase (pink color in the sketch) is denser and has a higher refractive index[105] than the PEG-rich phase (yellow), as can be observed in the bright-field images. The degree of vesicle osmotic deflation,r, is given by the ratio of the external to initial internal osmolarity. The lower panel shows snapshots vesicles trapped in a microfluidic device subjected to different deflation ratios. The dark regions visible in the bright-field images are the microfluidic posts. Scale bars: 5 μm. **b**–**e** FLIM phasor fluidity and dipolar relaxation

analysis for DOPC GUVs labeled with 0.5% LAURDAN in contact with the ATPS (**b**, **d**) and the glycinin condensates (**c**, **e**), respectively. The center of mass for fluidity changes (b, c) and dipolar relaxation (DR) changes (d, e) are shown. Histograms can be found in Supplementary Fig. 11. In both condensate systems, an increase in membrane wetting leads to a decrease in fluidity. Independent experiments are shown as circles and the lines represent mean values ± SD. The statistical analysis was performed with One-way ANOVA and Tukey post-test analysis ($p < 0.0001$, **** | $p < 0.001$, *** | $p < 0.01$, ** | $p < 0.05$, * | ns = non-significant). **f** Sketch summarizing the findings: when the condensate (pink) interacts with the membrane (green), wetting promotes an increased lipid packing (and decreased water dipolar relaxation) in the contact region. If the wetting affinity is increased, the effect on membrane packing becomes stronger. Note that the lipids in the contact region are colored in a darker green only for contrast, but the composition of the membrane remains unaltered. Data for panels (**b**-**e**) are provided as a Source Data file.

interface between the phase-separated polymer-rich phases[60,72] (Fig. 8a). These morphological changes are the product of the wetting transition of the polymer-rich condensate at the membrane[23,74], with stronger wetting as the deflation ratio r increases.

Figure 8b shows the changes in the fluidity fraction as measured by FLIM using the blue filter for DOPC GUVs encapsulating PEG/dextran ATPS. For this system, the analysis is performed taking into account the whole membrane, since we did not find significant differences for different membrane segments and because there is no phase separation in the first deflation step. Correlation between increased wetting and decrease in the fluidity fraction is observed. Note that for the first deflation step ($r = 1.1$), the membrane fluidity and dipolar relaxation do not change significantly compared to the initial state, but the effect is pronounced when phase separation occurs ($r \geq 1.3$); see Fig. 8b, d. For the PEG/dextran composition studied here, at deflation ratios above $r = 1.6$ the wetting affinity between the polymer-rich phases and the membrane remains the same, as previously shown by measurements of the intrinsic contact angle[72]. Thus,

we expect that for $r > 1.6$, the fluidity and dipolar relaxation fractions would remain unchanged. For comparison, Fig. 8c shows the results obtained for the glycinin condensates in contact with membranes obtained with FLIM, corresponding to the wetted membrane segment. These results corroborate the data obtained with spectral phasor analysis (Fig. 6) demonstrating the robustness of the method. Regarding the dipolar relaxation analysis obtained with FLIM using the green channel, Fig. 8d, e show the results for the ATPS and the glycinin condensates in contact with GUVs, respectively. For both systems, the water dipolar relaxations were reduced at increased wetting. These results indicate that water dynamics around LAURDAN are reduced for both analyzed systems when crowding and membrane wetting increases.

Using hyperspectral imaging and FLIM to analyze two different systems displaying changes in membrane wetting, we observed similar behavior regarding lipid packing and membrane hydration. Considering that the protein and the polymer systems form condensates with very different material properties[53], our results strongly suggest

that could be a universal feature for membranes interacting with bio-molecular condensates, as summarized in Fig. 8f.

## Discussion

Membrane wetting by biomolecular condensates has lately emerged as an exciting field because it is involved in diverse processes occurring in cell organization, development, and degradation[7]. Although membrane morphological transformations by condensate wetting transitions were first described many years ago[23], it was very recently that the biological relevance of membrane-condensate interactions came to the spotlight[7,8,28,75,76]. Understanding the intricate interactions between membranes and condensates necessitates the use of well-controlled in vitro models that allow pricise manipulation and monitoring of the physicochemical properties of both entities. In this direction, we recently performed a systematic investigation of membrane wetting by biomolecular condensates utilizing glycinin condensates in contact with GUVs. Our study demonstrated that fundamental factors such as salinity or membrane composition can tune their interaction[28]. In plant seeds, storage proteins like glycinin accumulate and undergo phase separation within vacuoles, contributing to the remodeling of vacuolar membranes during plant development[29,77] (see Supplementary Fig. 12). This interaction can lead to capillary-driven finger-like protrusions that can be reproduced in vitro[28,30]. While this phenomenon of vacuole remodeling has been observed for decades[30], it is only recently that we have been able to develop an experimental and theoretical framework for comprehending the wetting-driven remodeling processes[28]. It is important to highlight that the wetting transitions observed for glycinin in contact with membranes are not exclusive for this particular system but are instead a general feature observed across a range of condensate-membranes systems. Similar behavior has been wittnessed for very different systems, from phase-separated synthetic polymers to oligopeptide-rich coacervates[28,58]. Here, we resolve the mechanisms underpinning membrane-condensate interactions at a molecular scale using fluorescent microscopy combined with spectroscopic techniques. First, we use spectral phasor analysis of ACDAN fluorescence within condensates and show that ACDAN is sensitive to protein structural rearrangements via modification of the water environment surrounding the probe moiety. We prove that there are secondary structure rearrangements in the different stages of salt-driven phase separation of glycinin coacervates (Fig. 3a, b), revealing the responsiveness of these structures to slight changes in external conditions, such as salinity, pH, or temperature. Next, we used Raman spectroscopy and microscopy to show that the collective structure of water is modified in the condensate nano-environment. Raman spectroscopy has previously been used to determine protein concentration and to measure structural heterogeneity in single condensate droplets[78–80]. Our results indicate that Raman microscopy can also be used for monitoring structural changes that occur at different stages of the phase-separation process. Altogether, these results suggest a reciprocal mechanism whereby water activity can influence protein supramolecular rearrangement, while protein secondary structure can alter water dynamics in turn. Moreover, these changes in protein structure and water dynamics modulated by salinity are manifested as distinct mechanical and rheological properties of the condensates. (Fig. 4). Salt-dependent rheology has been previously shown for the P-granule protein, PGL-3[81]. Our results contribute to a deeper comprehension of the molecular origins of this behavior in glycinin condensates.

The crosstalk between protein structure and water dynamics is particularly interesting to discuss in the context of the intracellular milieu, where the high concentration of macromolecules and solutes can modify the physical properties of the cytoplasm, thereby influencing the molecular diffusion, signaling processes, and overall cellular physiology[34,35,82]. In this regard, the recently revisited "protoplasmic theory" of life provides a colloidal view of the cell physiology, in which the induction of cellular processes arise due to the close interaction between water and proteins[48,83,84]. Our results point to the same direction, suggesting a tight coupling between water activity and protein organization crucial to understanding phase separation in cell biology.

It is essential to highlight that ACDAN spectroscopic characteristics can supplement the current set of techniques[22,54] for measuring condensates properties. This is particularly relevant considering that most of the commercial confocal microscopes nowadays are capable of performing hyperspectral imaging[85].

In general, protein intrinsic fluorescence, normally governed by the fluorescence of tryptophan residues, could report on structural changes taking place at different conditions[86,87]. However, given the complexity of biomolecular condensates, instrinsic fluorescence is in general not suitable to evaluate changes occurring during phase separation, and extrinsic probes must be used for spectroscopic studies[87–89]. In particular, glycinin has 24 tryptophan residues, most of them hidden within the hydrophobic core, making autofluorescence unsuitable for evaluating the changes explored in this work (see Supplementary Fig. 13). It is also important to remark that the spectral properties of the fluorescent probes used here are not affected by the changes in sodium chloride concentration, as shown in Supplementary Fig. 14. In this manner, ACDAN has proven to be an excellent extrinsic reporter for the study of condensates, since the measurement of water dipolar relaxation, provides a fingerprint of a given state of the condensates. Indeed, it has been recently reported that ACDAN can distinguish between condensates formed by Dengue and Zika virus capsid proteins and RNA[12]. The combination of ACDAN fluorescence with the phasor approach could also constitute a quick and sensitive method for evaluating chemical changes within the condensates upon the addition of different components, like drugs[90] to modulate the phase separation process. Additionally, it could be applied for the prediction and the evaluation of biomolecules partitioning[91].

Membrane wetting by condesates can be regulated by tuning simple parameters like the salinity of the medium or the membrane composition[28]. Membranes have also been reported to regulate the assembly and nucleation of condensates[14,15]. By using LAURDAN hyperspectral imaging and FLIM together with the phasor analysis, we explored the membrane molecular changes induced by the wetting of biomolecular condensates. We prove that membranes in contact with condensates exhibit increased lipid packing and are more dehydrated than the bare condensate-free membranes (Fig. 6b). Previously, it was shown that macromolecular crowding induces membranes dehydration as a result of the reduced activity of water, even allowing the transition between different lyotropic phases[48]. This corroborates our finding that the highly crowded interface between a membrane in contact with a condensate, shows a more packed and dehydrated state than the bare membrane. Similarly, the hydration state of lipids has recently been shown to have a direct impact on membrane fluidity[92], and we and other groups have proven that the diffusion coefficient of the membrane in contact with condensates is slower than those in bare membranes[28,93]. Our results indicate that the slower diffusion observed in the membrane segment wetted by the condensate can be attributed to increased lipid packing and dehydration. It is noteworthy that the effect of polymers on membrane hydration has been extensively investigated for several decades. For example, during the 1980s and 1990s, the impact of PEG on membranes properties and lipid polymorphism was extensively studied. Lipid condensation effects and changes in the surface potential of phospholipid monolayers at air-water interfaces in the presence of PEG solutions were attributed to dehydration of the lipid polar headgroups[94]. Subsequently, it was shown that when added to lyposome suspensions, PEG promotes membrane dehydration due to long-range effects on water properties[95,96]. More recently, it was demonstrated that macromolecular crowding induces collective hydration of proteins over tens of angstroms[97]. These pieces of evidences accumulated over decades suggest that water plays an active role in biological

systems, mediating coupling mechanisms and structural rearrangements of macromolecules[48,83,84].

We employed FLIM with LAURDAN to distinguish between polarity/fluidity changes and water dipolar relaxation effects in the well-known PEG/dextran ATPS system. These polymers can be loaded into vesicles and be made to undergo liquid-liquid phase separation upon osmotic deflation of the vesicle(Fig. 8a). In this system, wetting between the polymer-rich phases and the membrane is modulated by sequential deflation. Again, we see that increasing wetting of the denser dextran-rich phase correlates with an increase in membrane lipid packing and dehydration (Fig. 8b), and that water activity also decreases (Fig. 8d). These results exhibit the same trend as that obtained for the protein condensate system (Fig. 8c, e). Altogether, our results suggest that changes in membrane hydration and lipid packing are a signature hallmark of membrane wetting by biomolecular condensates. The mechanism is summarized in Fig. 8f. This constitutes a convenient way by which cells could tune membrane packing via the wetting affinity of condensates. The sensitivity of FLIM with LAURDAN is not high enough to distinguish between membrane hemilayers, so we can only assume that observed changes occur in the hemilayer in contact with the condensate. In this regard, it has recently been reported that protein phase separation on membranes can promote transbilayer coupling[93]. This could give rise to domain-dependant signaling processes[98,99], providing a new mechanism for information transfer by coupling protein phase separation with membrane packing. In subsequent work, it would be interesting to explore the effect of wetting on membranes composed of complex lipid mixtures containing, for example, cholesterol to asses the influence of membrane phase state on wetting and the possible sorting of lipids in the region of interaction. Recently, using molecular dynamics simulations, negatively charged lipid sorting has been reported for polypeptide coacervates in contact with membranes[100], suggesting that this might also possible for the case of more complex protein condensates.

In this work, we provide a missing link describing the wetting of membranes by biomolecular condensates at the molecular scale. Undoubtedly, comprehending the nuances of how membrane interactions with condensates vary across different conditions is a complex task. Our results have demonstrated that changes in protein secondary structure at different salinity result in distinct and specific interactions with membranes. Future studies should continue to explore this direction in order to gain a comprehensive understanding of the crosstalk between membranes and biomolecular condensates.

## Methods
### Materials
1,2-dioleoyl-sn-glycero-3-phosphocholine (DOPC) was purchased from Avanti Polar Lipids (IL, USA). 6-acetyl-2-dimethylaminonaphthalene (ACDAN) was purchased from Santa Cruz Biotechnology (USA), and 6-dodecanoyl-2-dimethylaminonaphthalene (LAURDAN) from Thermofisher Scientific (USA). ATTO 647N-DOPE was obtained from ATTO-TEC GmbH (Siegen, Germany). Polydimethylsiloxane (PDMS) and curing agent were obtained as part of the SYLGARD® 184 silicone elastomer kit from Dow Corning (Michigan, USA). Dextran from Leuconostoc spp (Mw 450–650 kg/mol), and poly(ethylene glycol) (PEG 8000, Mw 8 kg/mol) were purchased from Sigma-Aldrich. Chloroform obtained from Merck (Darmstadt, Germany) was of HPLC grade (99.8%). The lipid stocks were mixed as chloroform solutions at 4 mM, containing 0.1 mol% ATTO 647N-DOPE or 0.5 mol % LAURDAN, and were stored until use at −20 °C. Fluorescein isothiocyanate isomer (FITC), sucrose, glucose, dimethyl sulfoxide (DMSO), sodium hydroxide (NaOH) and sodium chloride (NaCl) were obtained from Sigma-Aldrich (Missouri, USA). All aqueous solutions were prepared using ultrapure water from a SG water purification system (Ultrapure Integra UV plus, SG Wasseraufbereitung) with a resistivity of 18.2 MΩ cm.

### Vesicle preparation and deflation
DOPC giant unilamellar vesicles containing 0.5 mol % LAURDAN were prepared by electroformation[101]. Briefly, 2–4 μL lipid solution were spread onto indium tin oxide (ITO)-coated glasses and dried under vacuum for 1 h. The plates were assembled into a chamber with a Teflon spacer and the swelling solution (1.9 mL) was introduced. For electroformation, a sinusoidal electric field of 1.0 Vpp and 10 Hz was applied using a function generator for 1.5 h. For the studies with glycinin condensates, a sucrose solution was used for swelling, and once formed, the vesicles were diluted 1:1 in a glucose solution of the same osmolarity before use. In all cases, osmolarities of sucrose/glucose solutions matched the osmolarities of the condensate NaCl solutions (100–300 mOsm). The solution osmolarities were carefully adjusted using a freezing-point osmometer (Osmomat 3000, Gonotec, Germany).

The procedure for ATPS preparation and vesicle deflation together with the PEG/dextran phase diagram are described in detail by Zhao et al.[72]. Briefly, the swelling solution consisted of a mixture of dextran and PEG with initial weight ratio dextran:PEG = 1.57:1 (4.76% and 3.03% weight fractions, respectively). Afterwards, the GUVs containing ATPS were dispersed into an isotonic polymer solution with lower density (dextran:PEG 1:1, 3.54%, 3.54% weight fractions) to facilitate their sedimentation. Vesicles were loaded into a microfluidic chip. Deflation was controlled via a NeMESYS high-precision syringe pump (CETONI GmbH) by exchanging the external medium with a series of different hypertonic solutions containing constant polymer weight fractions and an increased weight fraction of sucrose.

### Microfluidics
The microfluidic device consists of a cascade GUV trapping system, which is described in detail elsewhere[72]. It was produced using PDMS precursor and curing agent (Sylgard 184, Dow Corning GmbH), at a mass ratio of 10:1. After polymerization at 80 °C for 2 h, inlet and outlet holes were punched with a biopsy punch with a plunger system (Kai Medical). Then the PDMS chips and glass coverslips were treated with plasma for 1 min using high-power expanded plasma cleaner (Harrick Plasma), before being bonded together. Before the experiments, the desired amount of solution was filled into the microfluidic device by centrifugation at 900xg (Rotina 420 R, Hettich). Solution exchange was performed with a NeMESYS high-precision syringe pump, at a flow speed of 1 μL/min for 40 min to ensure at least 10 fold exchange of the internal volume of the microfluidic device (~4 μL). For imaging, the flow speed was reduced to 0.035 μL/min to prevent vesicle movement.

### Protein extraction, purification, and labeling
Glycinin was purified as described by Chen et al.[31]. Briefly, defatted soy flour was dispersed 15-fold in water by weight and adjusted to pH 7.5 with 2 M NaOH. After centrifugation at 9000×g for 30 min at 4 °C, dry sodium bisulfite (SBS) was added to the supernatant (0.98 g SBS/L). The pH of the solution was adjusted to 6.4 with 2 M HCl, and the obtained turbid dispersion was kept at 4 °C overnight. Next, the dispersion was centrifuged at 6500×g for 30 min at 4 °C. The glycinin-rich precipitate was dispersed 5-fold in water, and the pH was adjusted to 7. The glycinin solution was then dialyzed against Millipore water for two days at 4 °C and then freeze-dried to acquire the final product with a purity of 97.5%[31].

To label the protein, 20 mg/mL soy glycinin solution was prepared in 0.1 M carbonate buffer (pH 9). A 4 mg/mL solution of FITC dissolved in DMSO was slowly added to the protein solution with gentle stirring to a final concentration of 0.2 mg/mL. The sample was incubated in the dark while stirring at 23 °C for three h. The excess dye was removed using a PD-10 Sephadex G-25 desalting column (GE Healthcare, IL, USA), and the buffer was exchanged with ultrapure water. The pH of the labeled protein solution was adjusted to 7.4 by adding 0.1 M NaOH. For fluorescence microscopy experiments, an aliquot of this solution was added to the working glycinin solution to a final concentration of 4%v/v.

For ACDAN hyperspectral imaging, the dye dissolved in DMSO was directly added to the unlabeled protein solution before the experiment to a final concentration of 5 μM.

## Formation of glycinin condensates

A 20 mg/mL glycinin solution at and pH 7 was freshly prepared in ultrapure water and filtered with 0.45 μm filters to remove any insoluble materials. To form the condensates, the desired volume of the glycinin solution was mixed with the same volume of a NaCl solution of twice the desired final concentration. In this manner, the final protein concentration was 10 mg/mL[28,31].

## Glycinin condensates in contact with membranes

The vesicle suspension was diluted 1:10 in a NaCl solution with the final NaCl concentration matching that of the condensates. The condensate suspension was diluted 1:4 and added to the vesicle suspension at a 15% v/v (condensate / GUV suspension) corresponding to a final condensate concentration of 0.4 mg/mL. After gently mixing the vesicle-condensate suspension, an aliquot of 100μL was placed on a coverslip (26×56 mm, Waldemar Knittel Glasbearbeitungs GmbH, Germany) for confocal microscopy and a chamber was formed using a round spacer and closing a coverslip. Coverslips were washed with ethanol and water before being passivated with a 10 mg/mL bovine serum albumin (BSA) solution.

## Hyperspectral imaging and fluorescence lifetime imaging microscopy (FLIM)

Hyperspectral and FLIM images were acquired using a confocal Leica SP8 FALCON microscope equipped with a 63×, 1.2 NA water immersion objective (Leica, Mannheim, Germany). The microscope was coupled to a pulsed Ti:Sapphire laser MaiTai (SpectraPhysics, USA), with a repetition rate of 80 MHz. A two-photon wavelength of 780 nm was used for ACDAN and LAURDAN excitation. Image acquisition was performed with a frame size of 512 × 512 pixels[2] and a pixel size of 72 nm. For hyperspectral imaging the xyλ configuration of the Leica SP8 was used, sequentially measuring in 32 channels with a bandwidth of 9.75 nm in the range from 416 to 728 nm. The FLIM data acquisition was performed by high-precision single-molecule detection hybrid detectors (HyD SMD, Leica) with GaAsP photocathodes. For the blue channel, the detection was set in the range of 416–470 nm, and for the green channel, 500–600 nm (see Supplementary Fig. 5). In all cases, 10–20 frames were accumulated. FLIM calibration of the system was performed by measuring the known lifetime of the fluorophore Coumarin 6 (100 nM) in ethanol (2.5 ns[102]). Hyperspectral and FLIM images were processed by the SimFCS software developed at the Laboratory of Fluorescence Dynamics, available on the webpage (https://www.lfd.uci.edu/globals/).

## Spectral phasor plot

LAURDAN and ACDAN fluorescence on hyperspectral imaging data were analyzed using the spectral phasor transform. This analysis calculates the real and imaginary component of the Fourier transform obtaining two quantities that are named G and S. The Cartesian coordinates (G,S) of the spectral phasor plot are defined by the following expressions:

$$G = \frac{\int_{\lambda_0}^{\lambda_f} I(\lambda) \cos(\omega n(\lambda - \lambda_0))d\lambda}{\int_{\lambda_0}^{\lambda_f} I(\lambda)d\lambda} \tag{1}$$

$$S = \frac{\int_{\lambda_0}^{\lambda_f} I(\lambda) \sin(\omega n(\lambda - \lambda_0))d\lambda}{\int_{\lambda_0}^{\lambda_f} I(\lambda)d\lambda} \tag{2}$$

where for a particular pixel represents the intensity as a function of wavelength, measured in the interval $(\lambda_0; \lambda_f)$. This range depends on

the used detector, in our case 416–728 nm. Note that changing the detection range will move the phasor position in the plot; the detection range must be conserved in order to be able to compare measurements. The parameter $n$ is the harmonic, i.e. the number of cycles of the trigonometric function that are fit in the wavelength range by means of the angular frequency $\nu$:

$$\nu = \frac{2\pi}{\lambda_f - \lambda_0} \tag{3}$$

In a real experiment we have a discrete number of spectral steps corresponding to the number of detection windows that cover the spectral range. For computational purposes, the spectral phasor transform expressed as a discrete transform in terms of the spectral channel is:

$$G = \frac{\sum_c^{N_c} I(c) \cos(2\pi c/N_c)}{\sum_c^{N_c} I(c)} \tag{4}$$

$$S = \frac{\sum_c^{N_c} I(c) \sin(2\pi c/N_c)}{\sum_c^{N_c} I(c)} \tag{5}$$

where $I(c)$ is the pixel intensity at channel $c$ and $N_c$ is the total number of channels. It is important that even if the number of spectral channels is small (in our case 32), the coordinates S and G are quasi continuous, since the photon counts in each pixel and channel $I(c)$ are high enough (~102) to allow a wide range of values in the coordinates S and G.

The spectral phasor position of a particular pixel carries information about the spectral intensity profile of that pixel. The spectral center of mass is related to the angle, while the distance from the center carries information on the spectrum broadness.

The spectral phasor approach follows the rules of vector algebra, known as the linear combination of phasors. This property implies that a combination of two independent fluorescent species will appear on the phasor plot at a position that is a linear combination of the phasor positions of the two independent spectral species. The fraction of each component is determined by the coefficients of the linear combination.

## Lifetime phasor plot

For the lifetime phasor plot, the fluorescence decay $I(\tau)$ was acquired at each pixel of an image and the coordinates were calculated and plotted according to:

$$G_{(\tau)} = \frac{\int_0^T I(\tau) \cos(\nu\tau)d\tau}{\int_0^T I(\tau)d\tau} \tag{6}$$

$$S_{(\tau)} = \frac{\int_0^T I(\tau) \sin(\nu\tau)d\tau}{\int_0^T I(\tau)d\tau} \tag{7}$$

where $\nu$ is the angular modulation frequency, and $\nu = 2\pi f$, where f is the laser repetition frequency and T is the period of the laser frequency. Note that the phase angle, $\varphi$ and the modulation, M can be obtained trough:

$$\varphi = \arctan(\nu\tau) \tag{8}$$

$$M = \frac{1}{\sqrt{1 + (\nu\tau)^2}} \tag{9}$$

In this manner, $G$ and $S$ can be expressed using the Weber notation:

$$G = M \cos \varphi \tag{10}$$

$$S = M \sin \varphi \tag{11}$$

## Two-cursor analysis

Exploiting the linear combination properties of the phasor plot[33], two-cursor analysis was used in all cases to calculate the histogram for the pixel distribution along the line (as shown in Fig. 1f) for dipolar relaxation changes of ACDAN and LAURDAN. When using the term fluidity obtained from LAURDAN fluorescence, we refer to changes in the order of the headgroup-chain interface[64], considering any process that can alter lipid rotational or translational rates. The histograms are presented as the number of pixels at each step along the line between two cursors, normalized by the total number of pixels. We plotted the average value for each histogram ± standard deviation, as well as the center of mass of the histogram for quantitative analysis with descriptive statistics. The center of mass was calculated as follows:

$$CM = \frac{\sum_{i=0}^{i=1} F_i \, i}{\sum_{i=0}^{i=1} F_i} \tag{12}$$

where $F_i$ is the fraction for fluidity or dipolar relaxation. Note that independently of the chosen position for the cursors in the phasor plot, the differences between the center of mass of the histograms will be determined statistically.

## Attenuated total reflectance-Fourier transform infrared spectroscopy (ATR-FTIR)

Spectra were acquired on a Vertex 70v spectrophotometer (Bruker Optik GmbH, Germany) equipped with a single reflection diamond reflectance accessory continuously purged with dried air to reduce water vapor distortions in the spectra. Samples (~3 μL) were spread on a diamond crystal surface, and dried under nitrogen flow to obtain the protein spectra. Sixtyfour accumulations were recorded at 25 °C using a nominal resolution of 4 cm⁻¹. Spectra were processed using Kinetic software developed by Dr. Erik Goormaghtigh at the Structure and Function of Membrane Biology Laboratory (Université Libre de Bruxelles, Belgium). After subtraction of water vapor and side chain contributions, the spectra were baseline corrected and the area normalized between 1700 and 1600 cm⁻¹. For better visualization of the overlapping components arising from the distinct structural elements, the spectra were deconvoluted using a Lorentzian deconvolution factor with a full width at the half maximum (FWHM) of 30 cm⁻¹ and a Gaussian apodization factor with a FWHM of 16.66 cm⁻¹ to achieve a line narrowing factor of $K = 1.8$[103]. In order to assign a determined band to the peaks in the spectra, a second derivative was performed on the Fourier self-deconvoluted spectra. The bands identified by both procedures were used as initial parameters for a least square iterative curve fitting of the original IR band ($K = 1$) in the amide I region, using mixed Gaussian/Lorentzian bands. Peak positions of each identified individual component were constrained within ±2 cm⁻¹ of the initial value. Details on the band assignment and the fitting can be found in Supplementary Table 1 and Supplementary Figs. 2 and 3.

## Raman microscopy

Raman images and spectra were acquired with a Raman confocal microscope Alpha300 R (WITec GmbH, Germany) with Zeiss EC Epiplan 50×/0.75 objective, at an excitation wavelength of 532 nm and 30 mW laser power. The image size was set to 15×15 μm² corresponding to 35×35 pixels². Spectra were acquired in the range 400–4100 cm⁻¹. The Raman band of the silicon wafer was used to calibrate the spectrometer. Data were analyzed with the Project FIVE v.5.2 data evaluation software from WITec. Spectral changes in the water band were quantified with the generalized polarization function $GP_{tetra/di}$ calculated from the intensity contributions of the bands at 3225 and 3432 cm⁻¹. These bands correspond respectively to tetra-coordinated and di-coordinated water molecules engaged in hydrogen bonds, as previously reported[48]:

$$GP_{tetra/di} = \frac{I_{3225} - I_{3432}}{I_{3225} + I_{3432}} \tag{13}$$

Here $I_{3225}$ and $I_{3432}$ represent the Raman intensities at the respective wavenumbers.

## FRAP measurements

FRAP measurements were performed on the SP8 setup equipped with a FRAP booster. The region of interest (ROI) was circular with a diameter of 2 μm. The condensates were bleached during 3 iterative pulses with a total time of ~3 s. Fluorescence intensities from ROIs corresponding to photobleaching were analyzed using ImageJ.

## Coalescence dynamics

The dynamics of condensate coalescence were recorded in brightfield mode by collecting 366 frames at a rate of 2.1 s/frame with a 63x/1.2NA water immersion objective. By fitting an ellipse to a pair of condensates of similar size, the aspect ratio was calculated as a/b, where a and b correspond to the long and short axes of the ellipse, respectively. The time evolution of the aspect ratio was fitted to the function $y = 1 + (y_0 - 1) \cdot \exp(-x/\tau)$, where $\tau$ is the characteristic relaxation time, and $y_0$ is the initial aspect ratio. The length scale "condensate size" in Fig. 4b corresponds to the final diameter of the condensates. The inverse capillary velocity $\frac{\eta}{\gamma}$, was obtained by plotting $\tau$ vs condensate size and by extracting the slope from the curves fitted to $y = \frac{\eta}{\gamma} x$.

## Rheology measurements

Bulk rheology of condensates was performed in oscillatory shear mode using an Anton-Paar MCR301 rheometer with 12 mm cone-plate (CP12) geometry. A solution of 10 mg/mL of glycinin at the desired salinity was prepared, with a final volume of 2 mL. Condensates were isolated by pelleting the phase-separated glycinin solution and the protein-rich phase was placed on the stage at 23 °C. Frequency sweeps were performed from 0.1–100 Hz up to 1% strain.

## Data reproducibility and statistics

At least three independent experiments were used to perform the statistical analysis. Histograms are shown as means ± standard deviation (SD). The center of mass are represented as scatter plots containing the individual measurements and the mean values ± SD. Results were analyzed using One-way ANOVA and Tukey post-test analysis ($p < 0.0001$, **** | $p < 0.001$, *** | $p < 0.01$, ** | $p < 0.05$, * | ns = non-significant). Statistical analyses and data processing were performed with the Origin Pro software (Originlab corporation). All microscopy images shown are representative of at least three independent experiments.

## Reporting summary

Further information on research design is available in the Nature Portfolio Reporting Summary linked to this article.

## Data availability

The source data underlying Figs. 1f, g, 2d, e, 3a–f, 4a–h, 6b–d, 8b–e as well as Supplementary Figs. 1a, b, 2, 3a, b, 4a–c, 5a–c, 8c–d, 9c, d,

10c–e, 11b, c, 13b–i, and 14a–f are provided in a separate Excel file labeled 'Source Data'. Source data are provided with this paper.

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

## Acknowledgements

A.M. acknowledges support from Alexander von Humboldt Foundation. Z.Z. acknowledges support from Free State of Thuringia (TAB; SARS-Rapid 2020-FGR-0051), and Deutsche Forschungsgemeinschaft (DFG, German Research Foundation) - project number 316213987-SFB 1278 (project D01). L.M. was supported in part by PEDECIBA, FOCEM—Fondo para la Convergencia Estructural del Mercosur (COF 03/11) and as Imaging Scientist by grant number 2020-225439 from the Chan Zuckerberg Initiative DAF, an advised fund of Silicon Valley Community Foundation. The authors would like to thank Dr. Nannan Chen for providing the purified protein, and Dr. Clemens Schmitt for the aid with the Raman experiments. We would like to express our sincere gratitude to the late Dr. Luis Bagatolli for his ideas and influence that have deeply inspired our work.

## Author contributions

A.M., L.M., and R.D. conceived the experiments and designed the project. R.D. supervised the project. A.M. performed most of the experiments. M.S. performed FTIR-ATR and intrinsic fluorescence experiments. N.W.T. performed the rheology experiments. Z.Z. aided with the ATPS systems and microfluidic setup. A.M., M.S., and L.M. analyzed the data. A.M., L.M., and R.D. wrote the paper, with input from the rest of the authors.

## Funding

## Competing interests

All authors declare no competing interests.
