## [Peer Review File · Nature Communications]

REVIEWER COMMENTS

Reviewer #1 (Remarks to the Author):

The manuscript titled "Biomolecular condensates modulate membrane lipid packing and hydration" by Mangiarotti et al. attempts to unravel the molecular mechanisms of membrane-condensate interactions as well as membrane wetting by biomolecular condensates, a biologically relevant process occurring in cell organization, development, and degradation. In this study, the authors have utilized fluorescence approaches such as phasor analysis of hyperspectral imaging and fluorescence lifetime imaging microscopy (FLIM) to study the molecular level changes occurring within phase-separated condensates of one of the most abundant storage proteins in the soybean, Glycinin. Using two nano-environmental sensitive probes, ACDAN and LAURDAN, the authors have characterized the water dynamics in the protein condensates at various salt concentrations and their interaction with membranes (GUVs). Further, authors have used FTIR-ATR to study the secondary structural changes and utilized vibrational Raman spectroscopy to comment on the structure and hydrogen bonding states of water occurring in the protein during different stages of the phase separation process. The authors build on their previous studies published in ACS Macro Letters (2020) and bioRxiv (<https://doi.org/10.1101/2022.06.03.494704>). This work highlights the utility of ACDAN and LAURDAN as excellent extrinsic reporters for the study of condensates that can potentially supplement the current set of techniques for studying condensate properties. I believe this manuscript will benefit the readers of this journal and support its publication. Overall, the manuscript is well-written although some typos need to be fixed. I recommend its publication in Nature Communications after a minor revision. Below I summarize my comments.

The authors used glycinin as a model phase-separating protein for this study. The choice of protein needs to be better justified. Glycinin is a highly abundant storage protein in the soybean and undergoes salt-dependent phase separation. What is not clear in the manuscript is why membrane-condensate interactions are important for glycinin (or for such a class of proteins). The authors can include the biological relevance of this class of proteins in the study of biomolecular condensates as well as membrane-condensate interactions.

The authors have used FTIR-ATR to monitor the secondary structural changes occurring within the condensates as a function of salt concentration and further studied the changes in the structure of water due to protein restructuring during phase separation. Do the authors think this modulation in the hydrogen bonding states of water within the condensates alters the material properties of condensates (liquid-like or gel-like) by performing relevant experiments like FRAP or FCS?

The authors have studied membrane hydration and lipid packing upon membrane-condensate interaction using LAURDAN fluorescence. They have shown the liquid phase condensation of glycinin in the vesicle interior upon a deflation. What do the authors observe/expect to see for deflation ratios > 1.6? Will the vesicle undergo fission and further decrease the fluidity fraction? Similarly, what happens to the fluidity fraction for NaCl concentration close to 200 mM where glycinin remains in the phase-separated state?

This group has previously shown that the structure and morphology of the soy protein condensates can be modulated by altering the solution conditions such as temperature and pH. A wide variety of driving forces are shown to be involved in the self-assembly of the protein during coacervation and condensation. Are these coacervates and hollow condensates expected to show similar membrane wetting and lipid packing changes upon interaction with GUVs?

The authors might consider moving water Raman spectra in condensates from Supplementary Figure 4 to the main figure (Figure 3).

The authors cited a conference abstract in Ref 64. They should replace this with the published paper(s) which described Raman spectroscopy of individual condensates (PNAS 2021: <https://doi.org/10.1073/pnas.2100968118> and Nature Communications 2022: <https://www.nature.com/articles/s41467-022-32143-0>).

Reviewer #2 (Remarks to the Author):

In the manuscript "Biomolecular condensates modulate membrane lipid packing and hydration" Mangiarotti et al report a study on biomolecular condensates and their interaction with membranes based on an ensemble of advanced microscopy techniques. Notably, they make use of ACDAN spectral imaging and phasor analysis (to detect relaxation due to water molecules), ATR-FTIR, Raman microscopy (spatially resolved analysis of water bands), FLIM of Laurdan (fluidity of the lipid bilayer).

The first part of the paper is focused on the analysis of Glycinin condensates. ACDAN imaging is sensitive to the crowded water environment inside the condensates. FTIR data show changes in secondary protein structure during phase separation. Raman imaging is used to reveal also changes in water structure (inside vs outside condensates). Overall, these data show elegantly, by a combination of different techniques, that condensation is associated with changes in secondary structures and water collective structure.

From a more methodological point of view, phasor analysis of ACDAN spectral imaging is a relatively new technique and these data show nicely its potential application for studying water in biomolecular condensates.

The second part of the paper is focused on the process of wetting (biomolecular condensate in contact with a GUV lipid vesicle). Here, Laurdan is used to reveal that fluidity of the membrane is modulated by interaction and state of the condensate (increasing salt  more wetting  less fluidity). This process is shown experimentally for two different systems (Glycinin cond.+GUV; PEG/Dextran inside vesicles) to reveal a general trend.

In my opinion, the paper is technically sound and of general interest and well worth of publication on Nat Commun.

I have only one major technical comment that I hope the authors can address. Besides that, I definitely recommend publication of this elegant work.

Major comments

1) In the second part of the paper, the authors study the wetting mechanism by comparing the Laurdan signal in the part of the membrane in contact with the condensate versus the part which is not in contact. I feel this measurement is somewhat delicate, since the resolution of the microscope is in the order of 200-300 nm, much larger than the membrane thickness.

In other words, is the signal from the membrane in contact with the condensate affected by signal from inside the condensate itself (even if it is a weak signal)?

The images suggest that the Laurdan signal inside the condensate is weak but it cannot be discerned if it is negligible or not.

How weak is the signal of Laurdan inside the condensate compared with the signal on the membrane?

(If this signal is not negligible, it could affect the measurements.)

In summary, I ask the authors to consolidate (with more explanation or analysis or controls...) that the signal comes exclusively from the membrane.

Minor comments

1) sup fig 4 panel d: the intensity should not be 'normalized' if the intensities are compared

Typo at page 17: orgin

Reviewer #3 (Remarks to the Author):

This manuscript presents experimental results documenting several physical properties within condensates and in "dry" membranes and membranes that are wet by protein or polymer condensates. Condensed protein or polymer drops are found to be dehydrated relative to the bulk and exhibit slower water dynamics. Similar measurements are conducted in membranes wet by condensates using LAURDAN combined with a spectral phasor analysis, and it is concluded that wet membranes are both dehydrated and more tightly packed than neighboring dry membranes.

This work appears to be carefully done. My main critique it is not clear that the authors can distinguish hydration effects from lipid packing effects in the LAURDAN measurements as stated. Addressing this may be a matter of a clearer explication, but may require additional controls. Also, the word "ordering" is used a lot and means different things to different people. A word with a more specific meaning tied to the results should be used instead. Maybe Headgroup-packing?

Specific comments: (roughly in order of appearance)

(first sentence of abstract) “critical phenomenon” means something different in thermodynamics than is usage here. This should be edited.

A quick google search produces decades old papers of FTIR studies of Glycinin (e.g. <https://pubs.acs.org/doi/10.1021/jf950340h>). Might be useful to emphasize what is new/unexpected about the current findings of Fig 3?

I find the presentation of the Laurdan results to be particularly confusing. I appreciate that this probe gives off different signals in the wet and dry regions of the membrane. I have more trouble assessing the validity of the interpretation that this contribution can be split cleanly into hydration and “membrane ordering” effects. Clearly, the protein/polymer environment in the condensate has altered polarity and water dynamics, as indicated in the data presented in the earlier figures, so this aspect of the Laurdan signal is expected. How do the authors gain confidence in also being able to independently detect lipid ordering using this probe? I am aware of the literature claiming this is possible, but has it been validated in a system like the one explored here? Is there a control that could make this interpretation more convincing to a non-expert in the details of this spectroscopy? At a minimum, the main text would greatly benefit from a clear explanation of why this experiment/analysis approach distinguishes these two important features. The current text refers to methods that did not clarify the issue for me. Second, the term “ordering” when it comes to lipids typically refers to chain ordering, which I do not think is being measured here. This also should be clarified.

There is a long history of membrane properties being studied as a function of hydration. It might be nice to frame the current results in this historical context. This is done a little in the discussion but could be done more. It seems straightforward that the membrane properties should be impacted by proximity to a wet protein/polymer drop – It would be more surprising if it were not impacted since it is a surface. This is not to say that the experimental result isn’t impactful, it would just be nice to put in the broader thermodynamic context.

I disagree with this statement in the discussion: “Our results show that the origin of the reduced fluidity and slower diffusion results from increased lipid packing and dehydration.” Why are these things causally linked? Aren’t these all consequences of their being different interactions in wet and dry regions?

The word "Remarkably" is used several times when referring to the direction measurements change with increased wetting. These results seem expected to me (more wetting = more interactions between membrane and condensate = more effected physical properties of the surface). Am I missing something? If I am understanding correctly, I think a different word should be used, e.g. "As expected"?)

REVIEWER COMMENTS

Reviewer #1

The manuscript titled "Biomolecular condensates modulate membrane lipid packing and hydration" by Mangiarotti et al. attempts to unravel the molecular mechanisms of membrane-condensate interactions as well as membrane wetting by biomolecular condensates, a biologically relevant process occurring in cell organization, development, and degradation. In this study, the authors have utilized fluorescence approaches such as phasor analysis of hyperspectral imaging and fluorescence lifetime imaging microscopy (FLIM) to study the molecular level changes occurring within phase-separated condensates of one of the most abundant storage proteins in the soybean, Glycinin. Using two nano-environmental sensitive probes, ACDAN and LAURDAN, the authors have characterized the water dynamics in the protein condensates at various salt concentrations and their interaction with membranes (GUVs). Further, authors have used FTIR-ATR to study the secondary structural changes and utilized vibrational Raman spectroscopy to comment on the structure and hydrogen bonding states of water occurring in the protein during different stages of the phase separation process. The authors build on their previous studies published in ACS Macro Letters (2020) and bioRxiv (<https://doi.org/10.1101/2022.06.03.494704>). This work highlights the utility of ACDAN and LAURDAN as excellent extrinsic reporters for the study of condensates that can potentially supplement the current set of techniques for studying condensate properties. I believe this manuscript will benefit the readers of this journal and support its publication. Overall, the manuscript is well-written although some typos need to be fixed. I recommend its publication in Nature Communications after a minor revision. Below I summarize my comments.

We thank the reviewer for correctly summarizing the main points of our work and for the positive consideration. We also appreciate the valuable comments and suggestions, which we address point by point below. We have performed new experiments, added a new section, and checked the typos. We sincerely believe the manuscript has improved thanks to the reviewer's constructive criticisms.

The authors used glycinin as a model phase-separating protein for this study. The choice of protein needs to be better justified. Glycinin is a highly abundant storage protein in the soybean and undergoes salt-dependent phase separation. What is not clear in the manuscript is why membrane-condensate interactions are important for glycinin (or for such a class of proteins). The authors can include the biological relevance of this class of proteins in the study of biomolecular condensates as well as membrane-condensate interactions.

We thank the reviewer for noticing this. We have now better explained the biological relevance of studying membrane-condensate interactions for this particular protein. We have added some paragraphs in the introduction and in the discussion sections as well as a new Supplementary Fig. 12 illustrating membrane remodeling by storage proteins in plant cells:

p.3 reads: "...The protein we use in this work to study phase separation and membrane wetting, glycinin, is one of the most abundant storage proteins in the soybean. Glycinin, along with other storage proteins, plays a crucial role in promoting vacuole membrane remodeling during embryogenesis in plants^{1, 2, 3}. Recent in vitro experiments have demonstrated its potential for membrane remodeling, highlighting its efficacy as a robust model for studying membrane-condensate interactions³..."

p.21 reads: "...In this direction, we recently performed a systematic investigation of membrane wetting by biomolecular condensates utilizing glycinin condensates in contact with GUVs. Our study demonstrated that fundamental factors such as salinity or membrane composition can tune their interaction³. In plant seeds, storage proteins like glycinin accumulate and undergo phase separation within vacuoles, contributing to the remodeling of vacuolar membranes during plant development^{1, 4} (see Supplementary Fig. 12). This interaction can lead to capillary-driven finger-like protrusions that can be reproduced in vitro^{2, 3}. While this phenomenon of vacuole remodeling has been observed for decades², it is only recently that we have been able to develop an experimental and theoretical

framework for comprehending the wetting-driven remodeling processes³. It is important to highlight that the wetting transitions observed for glycinin in contact with membranes are not exclusive for this particular system but are instead a general feature observed across a range of condensate-membranes systems. Similar behavior has been witnessed for very different systems, from phase-separated synthetic polymers to oligopeptide-rich coacervates^{3, 5, ...}”

In the included new Supplementary Fig. 12 we show that membrane remodeling processes by condensates in plants have been observed already in the 80s, but only after the recent introduction of the concept of condensates in cells and we can interpret those phenomena:

Supplementary Figure 12. **Wetting and remodeling of plant vacuolar membranes by storage protein condensates.** a-d. Electron micrographs of storage parenchyma cells of soybean cotyledons during development (p=protein droplets, v=vacuole): (a) protein droplets (red arrowheads) spread on the vacuolar membrane, (b, c) partially wet it, imposing additional curvature in the contact regions, and (d) forming “protein pockets” as finger-like structures (the red dotted line highlights the membrane contour) protruding towards the cytoplasm as a step prior to protein body formation. (e) Confocal microscopy image of protein storage vacuoles in *Arabidopsis thaliana* embryo cells: the protein droplets (red) wet the vacuolar membrane (green). Scale bars in (a-d) are 1 μ m and in (e) 10 μ m. Images (a-e) were adapted from reference ² with permission from SNCSC and image (f) from reference ¹.

The authors have used FTIR-ATR to monitor the secondary structural changes occurring within the condensates as a function of salt concentration and further studied the changes in the structure of water due to protein restructuring during phase separation. Do the authors think this modulation in the hydrogen bonding states of water within the condensates alters the material properties of condensates (liquid-like or gel-like) by performing relevant experiments like FRAP or FCS?

This is indeed a very interesting question, and we thank the reviewer for it. We have performed a series of experiments to address this, and built a new figure summarizing the results. Glycinin is a bulky protein, a hexamer of molecular weight of 360 kDa, which translates to the condensates’ high viscosity, on the same order of magnitude as that of the nucleolus^{3, 6, 7}. This precludes the measurements of material properties via FRAP or FCS, due to the slow diffusion of the proteins forming the condensate⁸. For that reason, to demonstrate the fluidity of the condensates, we performed (i) coalescence experiments, which allowed us to obtain the inverse capillary velocity, and (ii) rheology measurements to evaluate the condensates response to oscillatory stress. We see that indeed the material properties of condensates are salt-dependent, and there is a correlation with the changes we observed at molecular level. We have written a new section including a new figure (now Figure 4), two supporting figures (Supplementary Figs. 5 and 6), and include a paragraph in the discussion, see below:

p.10 reads:

“Condensate mechanical and rheological properties are tuned by salinity

Having proved that changing the salt concentration leads to rearrangements of the protein secondary structure and modifies the water nano-environment within condensates, we tested whether these

changes are reflected in the mechanical and rheological properties of the condensates. As indicated in the previous section, glycinin is a hexamer of high molecular weight (360kDa) that forms highly viscous condensates^{3, 8} with a viscosity on the order of 10^3 Pa.s. similar to that of the nucleolus^{6, 7}. Determining the diffusion coefficient of glycinin in the condensates using techniques like Fluorescence Recovery After Photobleaching (FRAP) is hindered by their high viscosity, as shown in Supplementary Fig. 5a. While the recovery curves exhibit reproducibility and discernable trend, these data do not allow quantifying the protein mobility within the condensates. Another conventional approach to obtain information regarding the material properties of condensates consists in monitoring their coalescence over time⁹. Glycinin condensates display coalescence within minutes⁸ (see Fig. 4a), with a relaxation time depending on the salinity, as shown in Fig. 4b. The slopes of the curves in Fig. 4b yield the inverse capillary velocity (η/γ), relating the viscosity (η) and surface tension (γ) of the condensates. As evidenced in Fig. 4c, the inverse capillary velocity changes for condensates at different salt concentrations, clearly indicating that the condensate material properties are dependent on the salinity. To further characterize the material properties of condensates, we performed bulk rheology measurements of the protein-rich phase at different salt conditions (see Methods). Figure 4d shows the change in the phase angle (δ) when varying frequency. The phase angle, defined as the tangent of the ratio between the loss and storage moduli, is a relative measure of the contributions of a material's viscous and elastic characteristics to its overall mechanical properties¹⁰: Purely elastic solids exhibit $\delta=0^\circ$ while purely viscous liquids have $\delta=90^\circ$. A plot of the phase angle at different frequencies provides an indication of the type of material under study and how it responds to different mechanical stresses (see Supplementary Fig. 6 for data interpretation). At all salt concentrations the condensate phase behaves as a viscoelastic liquid. Figures 4e-g show the storage (G') and loss (G'') moduli of the condensates, which together make up the complex shear modulus, $G^* = G' + iG''$, measured as functions of the oscillatory shear frequency for the salt concentrations under study. Quantities such as the complex viscosity (η^*) and the terminal relaxation time (τ_m) can be calculated from the complex modulus (see Supplementary Fig. 5b-c). The loss modulus, G'' , describing the viscous behavior of a material, will dominate at all frequencies for a purely viscous condensate, while the storage modulus, G' , will dominate for an elastic condensate⁹. In Figs. 4e-g, G'' dominates at short frequencies and long timescales, meaning the condensates behave more like liquids, while G' dominates at high frequencies and short timescales, with the condensates exhibiting more solid-like behavior. This general behavior is consistent across all the tested salt concentrations. From the linear part of the loss modulus in the low-frequency range, one can obtain the condensate phase zero-shear viscosity ($G'' = \omega\eta$, where ω is the frequency and η is the viscosity). Figure 4h show that the values of the obtained viscosities change considerably with salinity, and are in the order of kPa.s for 100 mM NaCl, which is consistent with values measured on individual glycinin droplets using microscopy approaches^{3, 8}. The frequency-dependent mechanical response displayed by glycinin condensates is similar to that of typical Maxwell fluids, presenting a single crossover point between the viscous and elastic regimes for the frequencies tested^{11, 12}. This crossover point occurs at different frequencies, depending on salt concentration, further indicating that the material properties of the condensates are changing with salinity. The terminal relaxation time, τ_m , can be calculated as the inverse of the crossover frequency, and indicates the average reconfiguration time of the protein network within the condensate. Supplementary Fig. 5c shows that the value of τ_m follows the order $\tau_m(50 \text{ mM}) > \tau_m(150 \text{ mM}) > \tau_m(100 \text{ mM NaCl})$. The obtained values are similar to those found for Arginine/Glycine-rich (R/G) peptides containing the RGRGG motif¹¹. Interestingly, this trend is similar to that observed for the inverse capillarity in Fig. 4c, and can be related also to the degree of water hydrogen bonding shown in Fig. 3e. Altogether, these results indicate that the salt-driven protein structural rearrangement that leads to changes in the water hydrogen bonding within the condensates results in altered mechanical and rheological properties of the condensates."

Figure 4. Glycinin condensates material properties change with salinity. The material properties of condensates at 50 mM (pink), 100 mM (green) and 150 mM (blue) NaCl were evaluated with different approaches. **a.** Examples of aspect ratio vs time for coalescing condensates of different sizes. Glycinin condensates coalesce within minutes, displaying different characteristic relaxation times according to size and NaCl concentration. The data is fitted with the function $y = 1 + (y_0 - 1) \cdot \exp(-x/\tau)$, where τ is the characteristic relaxation time. The inset shows an example of condensate coalescence at 100 mM NaCl. The scale bar is 5 μm . **b.** Plot of the relaxation time vs. the final condensate diameter. Solid lines are fits to the linear equation: $y = \frac{\eta}{\gamma} x$, where the slope, $\frac{\eta}{\gamma}$, is the inverse capillary velocity. **c.** Inverse capillary velocity obtained from (b) for varying salt concentrations indicate that the material properties of the condensates are modulated by salinity conditions. **d-h.** Rheology measurements of glycinin condensates at different salt concentrations display changes in the material properties. **d.** Phase angle vs frequency for glycinin condensates at the different conditions. In all cases condensates behave as viscoelastic liquids (see Supplementary Fig. 6 for data interpretation). **e-g.** Plots showing the average storage and loss modulus (G' , black, and G'' , red) vs frequency for glycinin condensates at the indicated salinities. Independent measurements are plotted as hollow circles and the mean \pm SD are shown in full circles ($n=3$). Shaded regions represent the the dominant viscous or elastic regime, as indicated. The crossover frequency (black dashed line) is equal to $1/\tau_m$, where τ_m is the terminal relaxation time. **h.** Zero-shear viscosity for the condensate phase at different salt concentrations. Individual data points are shown as circles and lines represent mean values \pm SD. Data for panels a-h are provided as a Source Data file.

Supplementary Figure 5: **Condensates material properties**. **a**. Fluorescence recovery after photo bleaching (FRAP) for glycinin condensates show negligible recovery, reflecting the high viscosity of these condensates, as previously reported⁸. Glycinin concentration is 10 mg/mL. Data show mean values and the shadowed area corresponds to the standard deviation ($n=4$ per condition). The insets show an example of a condensate at 100 mM NaCl before and after bleaching at the indicated times. Scale bar is 5 μm . **b**. Complex viscosity (η^*) vs frequency obtained by rheology measurements for the protein-rich (condensate) phase at different NaCl concentrations: 50 mM (pink), 100 mM (green), 150 mM (blue). Individual data points are shown as open circles and filled circles are mean \pm SD ($n=3$ per condition). **c**. Terminal relaxation time (τ_m) for the condensates at different NaCl concentrations. Individual data points are shown as circles, lines represent the mean \pm SD. Data for panels a-c are provided as a Source Data file.

Supplementary Figure 6: **Rheology measurements interpretation**. The sketches exemplify the typical responses of phase angle (δ) and storage (G') and loss (G'') moduli vs frequency for different types of materials. Adapted from reference ¹³.

p. 21 reads: “Altogether, these results suggest a reciprocal mechanism whereby water activity can influence protein supramolecular rearrangement, while protein secondary structure can alter water dynamics in turn. Moreover, these changes in protein structure and water dynamics modulated by salinity are manifested as distinct mechanical properties of the condensates. (Fig. 4). Salt-dependent rheology has been previously shown for the P-granule protein, PGL-3¹⁴. Our results contribute to a deeper comprehension of the molecular origins of this behavior in glycinin condensates.”

The authors have studied membrane hydration and lipid packing upon membrane-condensate interaction using LAURDAN fluorescence. They have shown the liquid phase condensation of glycinin in the vesicle interior upon a deflation. What do the authors observe/expect to see for deflation ratios > 1.6 ? Will the vesicle undergo fission and further decrease the fluidity fraction? The reviewer raises an interesting question, but we believe there must be a misunderstanding (or a typo). Our deflation experiments were not performed with glycinin condensates (outside the vesicles), but with condensates formed by the PEG/Dextran aqueous two-phase system (ATPS) encapsulated in the vesicles. In this latter case, when the deflation ratio (r) is above 1.6, for this particular PEG/Dextran composition, the intrinsic contact angle remains constant¹⁵ (see for example Figure 7 of reference 15). This implies that the wetting affinities of the polymer-rich phases to the membrane remain unaltered, and thus we expect the fluidity fraction to remain the same as for $r=1.6$. Figure S4 of the same reference¹⁵ shows results for the vesicle morphology at higher deflation ratios for this particular composition of PEG/Dextran (same as used in our current work). High deflation ratio results in producing more excess area, which is stored as nanotubes accumulating at the two-phase interface. The interfacial tension between the droplets provides the driving force for pinching them off and fissioning the two vesicle compartments (note that fissioning the membrane itself would require additional energy, which in cells is typically provided by constriction proteins such as dynamin or ESCRT). However, with further deflation, the accumulated membrane tubes at the interface act to lower the interfacial tension preventing the fission. It is important to note that this is the case when the deflation steps are done sequentially and slowly; at $r>1.6$ fission can be achieved by fast perturbations causing the vesicle to adopt a dumbbell morphology of two vesicles connected by a thin neck or a nanotube¹⁶. Since the morphological response is a little out of focus of the current work and covered in previous studies, we mainly address the question regarding the fluidity fraction. We have added now a paragraph in the text clarifying this:

p. 19 reads: “Note that for the first deflation step ($r=1.1$), the membrane fluidity and dipolar relaxation do not change significantly compared to the initial state, but the effect is pronounced when phase separation occurs ($r\geq 1.3$); see Fig. 8b,d. For the PEG/dextran composition studied here, at deflation ratios above $r=1.6$ the wetting affinity between the polymer-rich phases and the membrane remains the same, as previously shown by measurements of the intrinsic contact angle¹⁵. Thus, we expect that for $r>1.6$, the fluidity and dipolar relaxation fractions would remain the same.”

Similarly, what happens to the fluidity fraction for NaCl concentration close to 200 mM where glycinin remains in the phase-separated state?

We thank the reviewer for this question as it helped us realize a mistake in the display of Fig. 2a which is based on data from previous work³. At salt concentration of 200 mM, the droplets are already dissolved as shown in the corrected Fig. 2a. To address the question of the reviewer, we have observed that at 180 mM NaCl there is complete wetting of the membrane by glycinin condensates (i.e. the droplets spread completely on the membrane), and the interaction affinity is the strongest. The affinity is quantified by the geometric factor (Φ), which is a dimensionless parameter defined by the contact angles and their relation to the tensions of the membrane segments and the condensate interface³. As can be seen in the new Supplementary Fig. 8 included below, there is a small difference between the conditions of 150 mM and 180 mM: while for 150 mM $\Phi=-0.97$, for 180 mM $\Phi=-1$, i.e. $\Delta\Phi=0.03$. This change is very small compared to the differences observed between the other conditions: $\Delta\Phi=1.37$ between 50-100 mM and $\Delta\Phi=0.4$ between 100-150 mM. Thus, we expect that the fluidity fraction is slightly decreased for 180 mM, as a result of the increased interaction, but do not

expect a large difference compared to the fluidity fraction for 150 mM. We measured this new condition, and despite the trend observed in the histograms for the fluidity fraction, statistical analysis indicate no significant difference between 150 and 180 mM. These results are also discussed in the main text:

p. 15 reads: The strongest interaction between glycinin condensates and the membranes as determined by the geometric factor³ is observed at 180 mM NaCl (complete wetting, Supplementary Fig. 8b), however the difference in the fluidity fraction between this salt condition and 150 mM NaCl is negligible (Supplementary Figs. 8 c-d).

Supplementary Figure 8: **Geometric factor and interaction affinity and their relation to fluidity.** **a.** A sketch showing the three contact angles between the two membrane segments and droplet interface with the external solution. The contact angles and the respective tensions are related via the force balance triangle shown in the right. By measuring the angles with optical microscopy, the geometric factor (Φ) can be obtained as: $\Phi = (\sin \theta_e - \sin \theta_c) / \sin \theta_i$, for details see reference ³. This dimensionless factor is independent of the relative sizes of the droplet and vesicle and is determined by the material properties of the condensate and the membrane. The geometric factor provides an indirect measurement of the affinity contrast between the condensate and the membrane with respect to the external solution. **b.** Geometric factor for glycinin condensates in contact with vesicles at different NaCl concentrations. The system undergoes two wetting transitions, from dewetting ($[\text{NaCl}] = 43 \text{ mM}$) to partial wetting ($43 < [\text{NaCl}] < 180$) to complete wetting ($[\text{NaCl}] = 180 \text{ mM}$). **c.** Fluidity fraction histograms for vesicles in contact with glycinin condensates at the indicated NaCl concentrations. Data show mean \pm SD ($n=5$). **d.** Center of mass distribution for the histograms shown in c. There are not significant differences between the salt conditions of 150 and 180 mM NaCl. Individual data points are shown, the lines indicate mean \pm SD. Note that panels c and d show the same data as in Figure 6c, except for the composition of 180 mM NaCl. Panels a and b are adapted from reference ³. Data for panels c-d are provided as a Source Data file.

This group has previously shown that the structure and morphology of the soy protein condensates can be modulated by altering the solution conditions such as temperature and pH. A wide variety of driving forces are shown to be involved in the self-assembly of the protein during coacervation and condensation. Are these coacervates and hollow condensates expected to show similar membrane wetting and lipid packing changes upon interaction with GUVs?

We thank the reviewer for the interesting question. This is a good point, we speculate that the different triggers like pH, temperature or salinity can lead to condensates with different material properties and most likely different wetting affinity for membranes (as we have shown for the ionic strength). We investigated two very different systems, glycinin condensates and PEG/Dextran condensates, which have different viscosity and surface tension⁷, but they both have a similar effect on enhancing lipid packing. Thus, we would expect that the effect on lipid packing is universal for condensates wetting a membrane independently of the condensate chemistry, or the trigger (pH, temperature) causing phase separation.

To further prove this point and to answer the reviewer's question, we have performed a comparison between the isotropic ("regular") glycinin condensates and hollow condensates (new Supplementary Fig. 10, included below). In our previous work, we observed that hollow condensates can wet membranes and the wetting affinity can be tuned via salinity in a similar manner like for isotropic condensates³. Here, with the new experiments performed, we prove that hollow condensates also increase the lipid packing when comparing the wetted and the bare membrane (Supplementary Fig. 10c). In addition, when comparing the effect in packing between isotropic and hollow condensates at the same NaCl concentration (100 mM), we observed that hollow condensates produce a stronger effect on lipid packing. We speculated that this could be related to the existence of structural rearrangements in the protein network or secondary structure, leading to different material properties of the hollow condensates. When comparing the spectral response of ACDAN in hollow vs isotropic condensates, a reduced water dipolar relaxation is observed (Supplementary Fig. 10e). This result clearly suggest that the properties of hollow condensates are different from the isotropic ones at the same salinity condition, which explains the differences observed in lipid packing. This is indeed an interesting topic that requires a more detailed analysis in a future work.

Supplementary Figure 10: **Hollow condensates effect on membrane packing.** **a.** Hollow condensates can be formed exposing (isotropic) condensates to a sudden change in salinity to trigger phase separation within them, as indicated in the phase diagram sketch. This leads to the formation of a protein-poor phase (“hollow” void) surrounded by the protein-rich phase^{3, 8}. The images show glycinin condensates at 100 mM NaCl that become hollow after diluting the sample to 50 mM NaCl. **b.** Hollow condensates in contact with GUVs can wet and mold the membrane in a similar way as isotropic condensates. The confocal images are an example of a hollow condensate (green) wetting a GUV (red) at 150 mM NaCl. **c-e.** Histograms (upper panels) and center of mass (lower panels) obtained through hyperspectral imaging and phasor analysis of LAURDAN and ACDAN at 100 mM NaCl. Individual points are shown as circles and lines are mean±SD, n=5. **c.** Comparison between the bare and wetted segments of vesicles in contact with hollow condensates at 100 mM NaCl (final concentration in both cases). Similar to the behavior of isotropic condensates, the membrane wetted by the hollow condensate presents an increased packing compared to the bare membrane. Note that the fluidity fraction values obtained for the bare membrane correspond to those obtained for the bare membrane of GUVs in contact with isotropic condensates (compare to Fig. 6c). **d.** Comparison between the fluidity fraction for membranes in contact with isotropic or hollow condensates at 100 mM NaCl. The hollow condensates generate an increased membrane packing compared to isotropic ones at the same salinity. **e.** ACDAN shows a very different response for hollow condensates compared to isotropic ones at the same salinity. The lower dipolar relaxation observed for hollow condensates could imply differences in the protein secondary structure and hydrogen bonding. Figures a and b are adapted from reference³. All scale bars are 10 μm. Data for panels c-e are provided as a Source Data file.

The authors might consider moving water Raman spectra in condensates from Supplementary Figure 4 to the main figure (Figure 3).

We thank the reviewer for the suggestion. Now Figure 3 has been reorganized as follows:

Figure 3. Glycinin secondary structure changes with salt concentration and modifies the water environment inside condensates. **a.** Examples of FTIR-ATR spectra of the Amide I band of glycinin (10 mg/ml) in different regions of the phase diagram in Fig. 2a: R1 (0 mM NaCl), R2 (100 mM NaCl), R3 (400 mM NaCl); see Supplementary Figs. 2 and 3 for details. The inset shows a zoomed region highlighting the spectral shifts. **b.** Secondary structure content for glycinin at different conditions obtained by ATR-FTIR analysis. Individual data points are shown (circles) together with the mean \pm SD values (black lines). **c.** Percentage change in secondary structure motifs for the different salinity conditions relative to the structure of glycinin in salt-free water. The plotted data were obtained by subtracting the average values for each condition shown in b. The error bars were calculated as $\sigma_{i-j} = \sqrt{\sigma_i^2 + \sigma_j^2}$, where σ is the standard deviation. Major secondary structure rearrangements of the protein while changing salinity are associated with the α -helix and random+turns content. The most pronounced changes occur when glycinin enters the phase-separation region (R2) showing an increase in random coils and a decrease in α -helical structures. **d.** Raman microscopy image of a section of a single condensate at 100 mM NaCl. Pixel color is mapped to the intensity of the Amide I band (middle image) or water band (bottom image) as indicated by the color bar. The Amide I band increases and the water intensity decreases radially towards the interior of the condensate. Intensity profiles shown below the images were acquired along the white dashed lines in the images. Intensity profiles shown next to the images were acquired along the white dashed lines in the images. Scale bar is 3 μ m. **e.** Raman spectra of the water band at different NaCl concentrations. Lines are mean values and SD is shadowed ($n=3$). The regions in gray indicate the main bands around 3225 and 3432 cm^{-1} corresponding to tetra-coordinated and tri-coordinated water molecules respectively as shown by the cartoons. **f.** Spectral changes in the Raman water band quantified with the GP function ($GP_{tetra/di}$), calculated for the intensity contributions of the bands indicated in (e), see eq. 13. The observed changes suggest that the degree of hydrogen bonding of water is modified by the structural rearrangements of the protein at the different NaCl concentrations (see Supplementary Fig. 4 for further details). Individual measurements are plotted as circles and the lines represent mean \pm SD. Data for panels a-c and e-f are provided as a Source Data file.

The authors cited a conference abstract in Ref 64. They should replace this with the published paper(s) which described Raman spectroscopy of individual condensates (PNAS 2021: <https://doi.org/10.1073/pnas.2100968118> and Nature Communications 2022: <https://www.nature.com/articles/s41467-022-32143-0>).

We thank the reviewer for noticing this and we are sorry we have overlooked it. The issue is fixed now.

Reviewer #2:

In the manuscript "Biomolecular condensates modulate membrane lipid packing and hydration" Mangiarotti et al report a study on biomolecular condensates and their interaction with membranes based on an ensemble of advanced microscopy techniques. Notably, they make use of ACDAN spectral imaging and phasor analysis (to detect relaxation due to water molecules), ATR-FTIR, Raman microscopy (spatially resolved analysis of water bands), FLIM of Laurdan (fluidity of the lipid bilayer).

The first part of the paper is focused on the analysis of Glycinin condensates. ACDAN imaging is sensitive to the crowded water environment inside the condensates. FTIR data show changes in secondary protein structure during phase separation. Raman imaging is used to reveal also changes in water structure (inside vs outside condensates). Overall, these data show elegantly, by a combination of different techniques, that condensation is associated with changes in secondary structures and water collective structure.

From a more methodological point of view, phasor analysis of ACDAN spectral imaging is a relatively new technique and these data show nicely its potential application for studying water in biomolecular condensates.

The second part of the paper is focused on the process of wetting (biomolecular condensate in contact with a GUV lipid vesicle). Here, Laurdan is used to reveal that fluidity of the membrane is modulated by interaction and state of the condensate (increasing salt  more wetting  less fluidity). This process is shown experimentally for two different systems (Glycinin cond.+GUV; PEG/Dextran inside vesicles) to reveal a general trend.

In my opinion, the paper is technically sound and of general interest and well worth of publication on Nat Commun.

I have only one major technical comment that I hope the authors can address. Besides that, I definitely recommend publication of this elegant work.

We thank the reviewer for the very positive feedback and for considering our work sound and worth of publication in Nat Commun. We address the reviewer questions below and believe that the reviewer's comments have helped us improve the manuscript's clarity.

Major comments

1) In the second part of the paper, the authors study the wetting mechanism by comparing the Laurdan signal in the part of the membrane in contact with the condensate versus the part which is not in contact. I feel this measurement is somewhat delicate, since the resolution of the microscope is in the order of 200-300 nm, much larger than the membrane thickness.

In other words, is the signal from the membrane in contact with the condensate affected by signal from inside the condensate itself (even if it is a weak signal)?

The images suggest that the Laurdan signal inside the condensate is weak but it cannot be discerned if it is negligible or not.

How weak is the signal of Laurdan inside the condensate compared with the signal on the membrane?

(If this signal is not negligible, it could affect the measurements.)

In summary, I ask the authors to consolidate (with more explanation or analysis or controls...) that the signal comes exclusively from the membrane.

This is a very important point, and thanks to the reviewer, we realized that it needed further clarification in the manuscript. Considering this comment as well as the comments of reviewer 3, we have now re-written the section on LAURDAN measurements and included new figures to improve the clarity. In the first place, we would like to state that LAURDAN is a lipid-based probe and thus confined to the membrane. It has very poor solubility in water-based solutions and its fluorescence in aqueous environments is negligible (LAURDAN properties have been studied in depth by Parasassi and Gratton in several manuscripts). Thus, and we are certain that it is not partitioning to the condensates and is only located in the membrane, as we will explain below. In fact, we have tried to incorporate LAURDAN in the condensates as a control (without GUVs) and we could not succeed (no signal was detected in the condensates) even at micromolar concentrations (note that for vesicles we use nanomolar concentrations).

In the experiments using LAURDAN, first the vesicles are formed containing the dye, and afterwards the unlabeled condensates are added. In this way, the signal corresponding to LAURDAN is coming only from the membrane. In Figure 6b, it can be seen that the signal in the part of the membrane in contact with the condensate appears like a double line (see zoomed image in new Supplementary Figure 9, included below). This is an optical effect due to the high refractive index of the condensates, but the signal is coming from the labeled membrane. If we increase the contrast, we are able to detect protein autofluorescence coming from the condensates (see figure below). At the laser power we use (0.1% according to the SP8 microscope settings, within mW power) the signal from protein autofluorescence is very low, see Supplementary Fig 9b-c. In this case, we directly cut out the low intensity pixels from the histogram (see blue shadowed area) and we end up with the signal corresponding only to the membrane (note that we also discard the high intensity or saturated pixels).

Moreover, if there was fluorescence signal from LAURDAN in an environment different from that in the membrane, it should appear in a different position in the phasor plot. Because there is not bending of the LAURDAN trajectory from the membrane components, we can conclude that the data is associated with LAURDAN membrane fluorescence only.

One can also use masks to select the different parts of the membrane for analysis as shown in Figure 6b. Additionally, we have also measured the condensate autofluorescence in the absence of GUVs using the same settings as for the rest of the experiments. The histogram confirms that the signal is low and only a noise pattern is observed in the phasor plot (Supplementary Fig. 9d). In this manner, we are certain that we do not have any interfering signal in our measurements.

Indeed, and more importantly, if there was any signal coming from the condensate (i.e. strong autofluorescence from the protein), this would become evident in the phasor plot appearing as a third component in the data trajectory, and we would still be able to discriminate it. To exemplify this, we performed “proof-of-principle” experiments as explained below. When LAURDAN is in different membrane environments, it produces a linear trajectory in the spectral phasor plot, corresponding to the amount of water molecules around its moiety which is directly correlated with the lipid packing. Thus, we acquired hyperspectral images of DLPC vesicles and compared them to DOPC ones. Supplementary Fig. 7 (included below) shows that the combined DLPC and DOPC data display a linear trajectory. As DLPC melting temperature ($T_m = -2^\circ\text{C}$) is higher than that of DOPC ($T_m = -17^\circ\text{C}$), DLPC is more packed and less hydrated than DOPC, displaying a blue shift.

If we label the condensates with a water-soluble dye (here we used Sulforhodamine B, SRB), to enhance the signal from the condensates, and place them in contact with DOPC membranes labeled with LAURDAN, the data in the phasor plot no longer lies on the linear trajectory of LAURDAN but clearly displays a three-component behavior (Supplementary Fig. 7b below). The power of the phasor approach resides in the easy identification of processes taking place in the system and facile separation of contributions from the different components, without assuming a model *a priori*. Note that the classical approach of LAURDAN general polarization (GP) cannot distinguish such contributions, since it assumes a two state model¹⁷. In summary, we have proven that we do not have interference from other signals in our measurements. We have clarified this point now in the text:

p. 15 reads: “It is important to highlight that the measured fluorescence is only from LAURDAN present in the membrane segments, as shown in Supplementary Fig. 9, and the contribution from protein autofluorescence is negligible.”

Supplementary Figure 9: **Protein autofluorescence is negligible and does not interfere with LAURDAN measurements.** **a.** The image displayed in Figure 6b of a DOPC vesicle labeled with LAURDAN in contact with an unlabeled glycinin condensate at 100 mM NaCl, shows a double fluorescent line at the interface between the membrane and the condensate (see zoomed panel). This is due to an optical effect arising from the high refractive index of the condensates. **b.** LAURDAN fluorescence comes exclusively from the membrane, but when contrast is increased weak signal from protein autofluorescence can be detected from the condensate as exemplified in the zoomed panels. **c.** When analyzing the pixel intensity distribution with the phasor approach we can see that the autofluorescence contribution corresponds to low intensity pixels (shaded in blue), and appear as noise in the spectral phasor plot. We can eliminate these pixels and only analyze the pixels coming from the membrane (shaded in green), that appear as a coherent cloud in the phasor plot shown below the image. Note that we also eliminate high intensity and saturated pixels. **d.** The cutoff intensity in panel c is selected from measurements on condensates autofluorescence in the absence of vesicles and with the same setup used throughout the work (see Methods). Similar to panel c, the autofluorescence signal is very low and appears as noise in the phasor plot. Data for panels c-d are provided as a Source Data file.

Supplementary Figure 7: **LAURDAN fluorescence in different membranes describes a linear trajectory between various hydration states. a-b.** Proof-of-principle experiments showing the response of LAURDAN in membranes and how it changes when a third component is incorporated. **a.** DOPC and DLPC membranes labeled with LAURDAN (0.5 mol%) display a linear trajectory in the phasor plot, corresponding to different hydration and packing states. **b.** Fluorescence signal from probes other than LAURDAN do not appear along the linear trajectory (as displayed in panel a). This is exemplified with fluorescence signal from the condensates when labeled with the water-soluble dye Sulforhodamine B (SRB). In the phasor plot, it appears as a third component allowing its clear identification and separation of the signals, as exemplified here. For cases like this, a three cursor analysis would be required¹⁸ to unmix the signals and quantify the measured processes, in a similar way as the two-cursor analysis employed in this work.

Minor comments

1) sup fig 4 panel d: the intensity should not be 'normalized' if the intensities are compared
 We thank the reviewer for noticing this. The axis label of this panel (now Fig. 3e) was corrected.

Typo at page 17: orgin
 Corrected.

Reviewer #3 (Remarks to the Author):

This manuscript presents experimental results documenting several physical properties within condensates and in “dry” membranes and membranes that are wet by protein or polymer condensates. Condensed protein or polymer drops are found to be dehydrated relative to the bulk and exhibit slower water dynamics. Similar measurements are conducted in membranes wet by condensates using LAURDAN combined with a spectral phasor analysis, and it is concluded that wet membranes are both dehydrated and more tightly packed than neighboring dry membranes.

This work appears to be carefully done. My main critique it is not clear that the authors can distinguish hydration effects from lipid packing effects in the LAURDAN measurements as stated. Addressing this may be a matter of a clearer explication, but may require additional controls.

We thank the reviewer for the comments and suggestions, that made us note that the LAURDAN section was lacking clarity. We have now re-written the whole section and included new figures in the main text and supplementary file (all changes are indicated in blue). We answer the reviewer questions point-by-point, and we hope the manuscript is clearer now.

We would like to point out that LAURDAN is probably one of the most studied lipid fluorescent probes, and its photophysics in membranes has been extensively addressed in the literature and in particular by Dr. Malacrida. This was the main reason for choosing this particular dye. We have modified the manuscript including a more detailed introduction about LAURDAN photophysics and the information obtained from lipid membranes. We also explain in detail what we mean when using the term “fluidity”. The introduced changes and new figures are:

pp. 13-14 now read: “LAURDAN is a fluorescent probe sensitive to membrane polarity and water dipolar relaxation^{19, 20}. While the membrane polarity is related to the apparent dielectric constant at the LAURDAN location in the membrane, the dipolar relaxation corresponds to the re-orientation of the water molecules around the dye in response to the increase in LAURDAN’s dipole moment upon excitation (see Fig. 1b). The photophysics behind LAURDAN fluorescence in lipid membranes has been extensively described and reviewed for over 30 years^{19, 20, 21, 22}. In membranes, LAURDAN fluorescence is responsive to the dynamics of a few water molecules in the immediate environment of the bilayer, nearby the glycerol backbone of the glycerophospholipids¹⁹, as illustrated in Fig. 5. For this reason, LAURDAN has been widely used to assess the membrane phase state and hydration level^{19, 20, 23, 24, 25}. Lipid bilayers in the liquid phase (L_{α}) are less packed, more hydrated and with higher polarity compared to those in a gel (L_{β}) or liquid ordered (L_o) phase. They present greater dipolar relaxation, since the water molecules are able to reorient around the LAURDAN moiety during its excited state (see Fig. 1b). Membranes in the L_{β} or L_o phases are highly packed and dehydrated with low polarity, and the few water molecules present in the bilayer are not able to reorient while the dye is in the excited state. This is summarized in Fig. 5b, showing that LAURDAN fluorescence displays a big spectral shift (~50 nm) between the liquid phase and the gel phase. Between these extremes, there is a broad range of intermediate membrane hydration states that LAURDAN is sensitive to, and are related to different degrees of lipid packing. LAURDAN has been shown to be sensitive to small changes in membrane packing, even between lipids in the same phase state^{23, 24, 26, 27}. Here, we use the term “fluidity” referring to the membrane order parameters for a phospholipid. We consider that any process that affects the rotational or translational movement of lipids is changing the fluidity. The direct relationship between the LAURDAN spectral properties and the lipid order parameters was recently confirmed with nuclear magnetic resonance spectroscopy (H-NMR)²⁵.

The spectral phasor plot for the analysis of LAURDAN fluorescence has proven to be an outstanding and straightforward tool for the interpretation of the phenomena taking place at the membrane interface^{20, 26}. Due to the linear combination properties of the Fourier space, LAURDAN fluorescence in membranes produces a linear trajectory in the spectral phasor plot (see an example in Supplementary Fig. 7a), reflecting different packing and hydration states. The trajectory extremes correspond to the liquid and the gel phases^{20, 23, 24, 26, 27}, as illustrated in Fig. 5c.”

Figure 5. LAURDAN phasor analysis reports on membrane packing and hydration. **a.** Molecular structures of DOPC and LAURDAN. The dashed line indicates the approximate relative locations of the lipid and the dye from the bilayer center (~1.5 nm²⁸). **b.** Scheme illustrating the spectral shifts for LAURDAN in membranes with different properties: highly packed and dehydrated membranes, like those in the liquid-ordered (L_o) or in the gel-phase state (L_β) will present a blue-shifted spectrum with a maximum located near 440 nm. Membranes in a liquid phase state (L_α) will present a red shifted spectrum with a maximum centered around 490 nm^{19, 20}. **c.** Sketch of a spectral phasor plot showing the trajectory for LAURDAN fluorescence in membranes. Spectra corresponding to different degrees of water penetration will fall within the linear trajectory between the two extremes for the liquid and the gel phases^{20, 27, 29}. Any deviation from this trajectory would indicate the presence of a third component, as shown in Supplementary Fig. 7b.

Supplementary Figure 7: LAURDAN fluorescence in different membranes describes a linear trajectory between various hydration states. **a-b.** Proof-of-principle experiments showing the response of LAURDAN in membranes and how it changes when a third component is incorporated. **a.** DOPC and DLPC membranes labeled with LAURDAN (0.5 mol%) display a linear trajectory in the phasor plot, corresponding to different hydration and packing states. **b.** Fluorescence signal from probes other than LAURDAN do not appear along the linear trajectory (as displayed in panel a). This is exemplified with fluorescence signal from the condensates when labeled with the water-soluble dye Sulforhodamine B (SRB). In the phasor plot, it appears as a third component allowing its clear identification and separation of the signals, as exemplified here. For cases like this, a three cursor analysis would be required¹⁸ to unmix the signals and quantify the measured processes, in a similar way as the two-cursor analysis employed in this work.

We address the question about discriminating between hydration and packing effects in the specific comments below.

Also, the word “ordering” is used a lot and means different things to different people. A word with a more specific meaning tied to the results should be used instead. Maybe Headgroup-packing?

We agree, we have replaced the word “ordering” by “lipid packing”.

Specific comments: (roughly in order of appearance)

(first sentence of abstract) “critical phenomenon” means something different in thermodynamics than its usage here. This should be edited.

We agree and have replaced the word “critical” by “key”.

A quick google search produces decades old papers of FTIR studies of Glycinin (e.g. <https://pubs.acs.org/doi/10.1021/jf950340h>). Might be useful to emphasize what is new/unexpected about the current findings of Fig 3?

We thank the reviewer for pointing this out, and we have now clarified this in the main text. Indeed, glycinin is a protein that has been very well studied because it has applications in several fields, from materials science to food science. While all the literature available evaluates the structure of glycinin in water (as a soluble protein), in our work we evaluated the FTIR at different points in the phase diagram, including conditions at which the protein forms condensates. To the best of our knowledge, FTIR spectra for the different phases in the phase diagram have not been reported, which is a novel aspect of our work. We have clarified this in the manuscript:

p. 8 reads: “..Widely used in material and food sciences, glycinin solutions in water have been extensively studied using FTIR-ATR³⁰. Here, we report the secondary structural changes of glycinin in various conditions, under which phase separation occurs.”

I find the presentation of the Laurdan results to be particularly confusing. I appreciate that this probe gives off different signals in the wet and dry regions of the membrane. I have more trouble assessing the validity of the interpretation that this contribution can be split cleanly into hydration and “membrane ordering” effects. Clearly, the protein/polymer environment in the condensate has altered polarity and water dynamics, as indicated in the data presented in the earlier figures, so this aspect of the Laurdan signal is expected. How do the authors gain confidence in also being able to independently detect lipid ordering using this probe? I am aware of the literature claiming this is possible, but has it been validated in a system like the one explored here? Is there a control that could make this interpretation more convincing to a non-expert in the details of this spectroscopy? At a minimum, the main text would greatly benefit from a clear explanation of why this experiment/analysis approach distinguishes these two important features. The current text refers to methods that did not clarify the issue for me. Second, the term “ordering” when it comes to lipids typically refers to chain ordering, which I do not think is being measured here. This also should be clarified.

We agree with the last part of this comment for the term “ordering” (no longer used) and hope the revisions and the new figures we have included introducing LAURDAN photophysics clarify this point.

With respect to what LAURDAN measures and how to split the dipolar relaxation from the polarity changes, we agree with the reviewer in that this was not clear in the previous version of the manuscript, and now we have expanded the explanations in the FLIM section.

We would like to emphasize that in our experimental system, LAURDAN is located exclusively on the membrane, and the changes we are measuring report changes in membrane properties. This is now better explained in the manuscript (see also the answer to reviewer 2). Regarding the polarity/packing and dipolar relaxation changes explored by hyperspectral imaging, both variables are measured together. However, they can be split using a proper set of filters with time-resolved spectroscopy, like FLIM. The two measurements (dipolar relaxation and polarity) are supported by different physical variables. The first by the re-orientation of water molecules around the probe, and the second by the apparent dielectric constant. In the blue channel, the lifetime measures the polarity because there is no relaxation of water at this part of the spectrum (relaxation processes correspond to lower energies, *i.e.* longer wavelengths). However, in the green channel we can quantify the occurrence of water relaxation. These confined water molecules have a slower rotational time (ns) than those in bulk (ps), and the pixel cloud is observed at the FLIM phasor plot outside the universal

semicircle. This is because the time required for water relaxation (ns) compete with the fluorescence (ns) resulting in phase delay in the phasor plot pulling the data outside the universal circle. This constitutes an unequivocal measurement of the water relaxation. Now we have explained this in detail and included a new figure to make this point clearer:

pp. 17-18 read: "FLIM measurements involve exciting the sample with a pulsed laser, and recording the emission intensity vs time at each pixel. The excited fluorophores will give rise to a modulated emission shifted in phase relative to the exciting light³³. The resulting lifetime information can be represented in a phasor plot, as shown in Fig. 7a. The semicircle is called the universal semicircle, and all mono-exponential lifetimes will fall within it³³. The modulation (M) represents the distance of the phasor point from the origin, while the phase angle (φ) determines the position of the phasor point in the graph (Fig. 7a, equations 8-11). The lifetimes (τ) increase counter-clockwise; when φ increases and M decreases, the lifetime is longer. Figure 7b shows that processes occurring during the excited state of the dye, such as dipolar relaxations, Förster resonance energy transfer (FRET) or excimer formation, can be easily identified because the phasor data will appear outside the universal semicircle^{33, 43, 72}. This results from a delay in the emission due to the time required for these processes to take place. In the case of dipolar relaxations, the delay is caused by the time required to reorganize water molecules around the LAURDAN excited dipole (see Fig. 1b)^{33, 43, 72}. The additional phase shift in the excited state of the probe, results in an overall rotation of the plot moving the phasor distribution outside the universal semicircle^{33, 70} (Fig. 7b). Using a "blue" filter (*i.e.* collecting the emission at 416-470 nm) allows measuring changes in polarity due to changes in the apparent dielectric constant (Fig. 7c), while with a "green" filter (500-600 nm) we can isolate the dipolar relaxation contributions to the lifetime^{33, 39, 43, 70}(Fig. 7d, Supplementary Fig. 11). All properties of the Fourier space described for the spectral phasors, such as the linear combinations and the reciprocity principle, also apply here, allowing the quantification of the observed changes (see Fig. 1)."

Figure 7. Lifetime phasor plots allow discriminating between polarity and dipolar relaxation changes for LAURDAN decay in membranes. **a.** Lifetime phasor plot. The modulation (M) indicates the distance of the phasor point from the origin (0:0), and the phase angle (φ) the decrease or increase in lifetime (τ). Together, these

parameters determine the position of the phasor point in the universal semicircle. **b.** When excited-state processes take place, M remains the same, but due to the delay in the emission ($\Delta\phi$), the phasor points appear outside the universal circle as the plot rotates^{33, 43, 70, 72}. **c-d.** Using different bandpass filters, the contributions of the polarity and the dipolar relaxation can be split for LAURDAN decay. **c.** The lifetimes measured through the blue channel (416-470 nm) give information about the change in lipid packing. The sketch illustrates how the lifetime changes between liquid and gel phases. Linear combination rules apply and the changes can be quantified in the same manner as described for spectral phasors (Fig. 1). All intermediate packing states will fall within the linear trajectory between these two extremes. **d.** The green channel (500-600 nm) for LAURDAN fluorescence provides information on water dipolar relaxation processes, and the phasor points fall outside the universal semicircle. For membranes in the liquid phase state, dipolar relaxations are more pronounced because the water molecules have enough time to reorient around the LAURDAN moiety (see Fig. 1b), while for gel and liquid ordered phases dipolar relaxation are less pronounced.

Note that Supplementary Fig. 11 show the data for FLIM obtained in the blue and green channels. The fact that in the green channel the pixel cloud is located outside the universal semicircle clearly indicates that we are measuring an excited-state process, namely the water dipolar relaxation^{29, 31, 33}.

There is a long history of membrane properties being studied as a function of hydration. It might be nice to frame the current results in this historical context. This is done a little in the discussion but could be done more. It seems straightforward that the membrane properties should be impacted by proximity to a wet protein/polymer drop – It would be more surprising if it were not impacted since it is a surface. This is not to say that the experimental result isn't impactful, it would just be nice to put in the broader thermodynamic context.

We appreciate the reviewer's feedback and agree with their observation. Indeed, the impact of hydration effects resulting from polymer interaction with membranes has been documented for many years. In our revised manuscript, we have expanded the discussion to delve deeper into this aspect and provide additional insights into the subject matter.

I disagree with this statement in the discussion: "Our results show that the origin of the reduced fluidity and slower diffusion results from increased lipid packing and dehydration." Why are these things causally linked? Aren't these all consequences of their being different interactions in wet and dry regions?

We agree with the reviewer in that the phrase was not correctly stated. However, we observe a correlation between increased packing, dehydration and reduced lipid mobility. In particular, when comparing the wetted and bare membrane we see a marked decrease in diffusion for the wetted segment³. As the reviewer indicated, this difference is clearly due to the presence of the condensate in the wetted membrane segment. The membrane changes that we measured here indicate that membrane packing and hydration are higher in the membrane in contact with the condensate, indicating that the interaction leads to these changes in the membrane. This conclusion is reinforced in Figure 6c and Figure 8 in which we are measuring how the wetted part of the membrane becomes more packed when the interaction is stronger (via increasing the NaCl concentration for the glycinin condensates or the deflation ratio for the PEG/Dextran ATPS). An increased lipid packing and dehydration are directly related with a slower lipid diffusion, as recently proved by directly changing the humidity of the bilayer and measuring the impact on lipid mobility³⁴.

p. 23 now reads: "Our results indicate that the slower diffusion observed in the membrane segment wetted by the condensate can be attributed to increased lipid packing and dehydration."

The word "Remarkably" is used several times when referring to the direction measurements change with increased wetting. These results seem expected to me (more wetting = more interactions between membrane and condensate = more effected physical properties of the surface). Am I missing something? If I am understanding correctly, I think a different word should be used, e.g. "As expected"?)

The word "remarkably" was used on three occasions. It is not straightforward to expect that wetting by the condensates will increase the packing of the lipids. On the contrary, another

intuitive picture would be a decreasing packing due to protein residues intercalating between the lipids. We have reworded the manuscript and reduced the use of the word “remarkably”.

References:

1. Feeney M, Kittelmann M, Menassa R, Hawes C, Frigerio L. Protein Storage Vacuoles Originate from Remodeled Preexisting Vacuoles in *Arabidopsis thaliana*. *Plant Physiology* **177**, 241-254 (2018).
2. Yoo BY, Chrispeels MJ. The origin of protein bodies in developing soybean cotyledons: a proposal. *Protoplasma* **103**, 201-204 (1980).
3. Mangiarotti A, Chen N, Zhao Z, Lipowsky R, Dimova R. Wetting and complex remodeling of membranes by biomolecular condensates. *Nature Communications* **14**, 2809 (2023).
4. Herman EM, Larkins BA. Protein Storage Bodies and Vacuoles. *The Plant Cell* **11**, 601-613 (1999).
5. Lu T, *et al.* Endocytosis of Coacervates into Liposomes. *Journal of the American Chemical Society* **144**, 13451-13455 (2022).
6. Brangwynne CP, Mitchison TJ, Hyman AA. Active liquid-like behavior of nucleoli determines their size and shape in *Xenopus laevis* oocytes. *Proceedings of the National Academy of Sciences of the United States of America* **108**, 4334-4339 (2011).
7. Wang H, Kelley FM, Milovanovic D, Schuster BS, Shi Z. Surface tension and viscosity of protein condensates quantified by micropipette aspiration. *Biophysical Reports* **1**, 100011 (2021).
8. Chen N, Zhao Z, Wang Y, Dimova R. Resolving the Mechanisms of Soy Glycinin Self-Coacervation and Hollow-Condensate Formation. *ACS Macro Letters* **9**, 1844-1852 (2020).
9. Alshareedah I, Kaur T, Banerjee PR. Chapter Six - Methods for characterizing the material properties of biomolecular condensates. In: *Methods in Enzymology* (ed Keating CD). Academic Press (2021).
10. Carreau PJ, De Kee DCR, Chhabra RP. Rheology of Polymeric Systems. In: *Rheology of Polymeric Systems (Second Edition)* (eds Carreau PJ, De Kee DCR, Chhabra RP). Hanser (2021).
11. Alshareedah I, Moosa MM, Pham M, Potoyan DA, Banerjee PR. Programmable viscoelasticity in protein-RNA condensates with disordered sticker-spacer polypeptides. *Nature Communications* **12**, 6620 (2021).
12. Jawerth L, *et al.* Protein condensates as aging Maxwell fluids. *Science* **370**, 1317-1323 (2020).
13. Ramli H, Zainal NFA, Hess M, Chan CH. Basic principle and good practices of rheology for polymers for teachers and beginners. **4**, 307-326 (2022).
14. Jawerth LM, *et al.* Salt-Dependent Rheology and Surface Tension of Protein Condensates Using Optical Traps. *Physical Review Letters* **121**, 258101 (2018).
15. Zhao Z, Roy D, Steinkühler J, Robinson T, Lipowsky R, Dimova R. Super-resolution imaging of highly curved membrane structures in giant vesicles encapsulating molecular condensates. *Advanced Materials* **34**, 2106633 (2021).
16. Dimova R, Lipowsky R. Giant Vesicles Exposed to Aqueous Two-Phase Systems: Membrane Wetting, Budding Processes, and Spontaneous Tubulation. *Advanced Materials Interfaces* **4**, 1600451 (2017).
17. Malacrida L, Gratton E, Jameson DM. Model-free methods to study membrane environmental probes: a comparison of the spectral phasor and generalized polarization approaches. *Methods and Applications in Fluorescence* **3**, 047001 (2015).
18. Sameni S, Malacrida L, Tan Z, Digman MA. Alteration in Fluidity of Cell Plasma Membrane in Huntington Disease Revealed by Spectral Phasor Analysis. *Scientific Reports* **8**, 734 (2018).
19. Bagatolli LA. LAURDAN Fluorescence Properties in Membranes: A Journey from the Fluorometer to the Microscope. In: *Fluorescent Methods to Study Biological Membranes* (eds Mély Y, Duportail G). Springer Berlin Heidelberg (2013).
20. Gunther G, Malacrida L, Jameson DM, Gratton E, Sánchez SA. LAURDAN since Weber: The Quest for Visualizing Membrane Heterogeneity. *Accounts of Chemical Research* **54**, 976-987 (2021).
21. Parasassi T, De Stasio G, Ravagnan G, Rusch RM, Gratton E. Quantitation of lipid phases in phospholipid vesicles by the generalized polarization of Laurdan fluorescence. *Biophysical Journal* **60**, 179-189 (1991).
22. Parasassi T, De Stasio G, d'Ubaldo A, Gratton E. Phase fluctuation in phospholipid membranes revealed by Laurdan fluorescence. *Biophysical Journal* **57**, 1179-1186 (1990).
23. Mangiarotti A, Bagatolli LA. Impact of macromolecular crowding on the mesomorphic behavior of lipid self-assemblies. *Biochimica et Biophysica Acta (BBA) - Biomembranes* **1863**, 183728 (2021).
24. Malacrida L, Astrada S, Briva A, Bollati-Fogolin M, Gratton E, Bagatolli LA. Spectral phasor analysis of LAURDAN fluorescence in live A549 lung cells to study the hydration and time evolution of intracellular lamellar body-like structures. *Biochimica et Biophysica Acta (BBA) - Biomembranes* **1858**, 2625-2635 (2016).

25. Leung SSW, Brewer J, Bagatolli LA, Thewalt JL. Measuring molecular order for lipid membrane phase studies: Linear relationship between Laurdan generalized polarization and deuterium NMR order parameter. *Biochimica et Biophysica Acta (BBA) - Biomembranes* **1861**, 183053 (2019).
26. Golfetto O, Hinde E, Gratton E. The Laurdan Spectral Phasor Method to Explore Membrane Microheterogeneity and Lipid Domains in Live Cells. In: *Methods in Membrane Lipids* (ed Owen DM). Springer New York (2015).
27. Gratton E, Digman MA. Laurdan identifies different lipid membranes in eukaryotic cells. In: *Cell Membrane Nanodomains: from Biochemistry to Nanoscopy* (eds Cambi A, Lidke DS). CRC Press (2014).
28. Jurkiewicz P, Olżyńska A, Langner M, Hof M. Headgroup Hydration and Mobility of DOTAP/DOPC Bilayers: A Fluorescence Solvent Relaxation Study. *Langmuir* **22**, 8741-8749 (2006).
29. Malacrida L, Ranjit S, Jameson DM, Gratton E. The Phasor Plot: A Universal Circle to Advance Fluorescence Lifetime Analysis and Interpretation. *Annual Review of Biophysics* **50**, 575-593 (2021).
30. Long G, Ji Y, Pan H, Sun Z, Li Y, Qin G. Characterization of Thermal Denaturation Structure and Morphology of Soy Glycinin by FTIR and SEM. *International Journal of Food Properties* **18**, 763-774 (2015).
31. Ranjit S, Malacrida L, Jameson DM, Gratton E. Fit-free analysis of fluorescence lifetime imaging data using the phasor approach. *Nature Protocols* **13**, 1979-2004 (2018).
32. Štefl M, James NG, Ross JA, Jameson DM. Applications of phasors to in vitro time-resolved fluorescence measurements. *Analytical Biochemistry* **410**, 62-69 (2011).
33. Golfetto O, Hinde E, Gratton E. Laurdan Fluorescence Lifetime Discriminates Cholesterol Content from Changes in Fluidity in Living Cell Membranes. *Biophysical Journal* **104**, 1238-1247 (2013).
34. Chattopadhyay M, Krok E, Orlikowska H, Schwille P, Franquelim HG, Piatkowski L. Hydration Layer of Only a Few Molecules Controls Lipid Mobility in Biomimetic Membranes. *Journal of the American Chemical Society* **143**, 14551-14562 (2021).

REVIEWERS' COMMENTS

Reviewer #1 (Remarks to the Author):

The authors have addressed all of my concerns and the manuscript can now be accepted for publication in Nature Communications.

Reviewer #3 (Remarks to the Author):

The revised manuscript addresses many of my comments.

To follow up on the issue regarding “order parameter” or “order” – the paper cited (ref 65) that shows a correlation between laurdan GP and “NMR order parameters” shows that a linear correlation exists for the methylene at the headgroup-chain interface (SCD,max), but the same correlation is not shown for other segments of the chain, or for some average value representing the chain overall. (I am not sure why this isn't discussed since the information is in the acquired spectra, maybe because the correlation isn't as nice?) The abstract of this paper does make this point clear: “This observed correlation supports the idea that lipid chain order is tightly associated with the amount and dynamics of water molecules at the glycerol backbone level of the membrane.”

It certainly makes a lot of sense that the most ordered methylene would be most reflective of the LAURDAN GP, since this is where the fluorophore is generally located. The current manuscript uses this reference to justify the statement that the Laurdan signal represents the chain order parameter, which is typically used to describe chain anisotropy all along the chain (often an average over all chain segments). I recommend further tightening up this language to clarify what can directly concluded. (This has been done pretty carefully already, but there are a few places, e.g. the 2 places where fluidity is defined) My recommendation would be a word or phrase that indicates the observations reflect the ordering of the headgroup-chain interface, or such as headgroup packing, or something similar (which is language used elsewhere in the text).

REVIEWERS' COMMENTS

Reviewer #1 (Remarks to the Author):

The authors have addressed all of my concerns and the manuscript can now be accepted for publication in Nature Communications.

We thank the reviewer for all the comments and suggestions that help us improve our work.

Reviewer #3 (Remarks to the Author):

The revised manuscript addresses many of my comments.

To follow up on the issue regarding “order parameter” or “order” – the paper cited (ref 65) that shows a correlation between laurdan GP and “NMR order parameters” shows that a linear correlation exists for the methylene at the headgroup-chain interface (SCD,max), but the same correlation is not shown for other segments of the chain, or for some average value representing the chain overall. (I am not sure why this isn't discussed since the information is in the acquired spectra, maybe because the correlation isn't as nice?) The abstract of this paper does make this point clear: “This observed correlation supports the idea that lipid chain order is tightly associated with the amount and dynamics of water molecules at the glycerol backbone level of the membrane.”

It certainly makes a lot of sense that the most ordered methylene would be most reflective of the LAURDAN GP, since this is where the fluorophore is generally located. The current manuscript uses this reference to justify the statement that the Laurdan signal represents the chain order parameter, which is typically used to describe chain anisotropy all along the chain (often an average over all chain segments). I recommend further tightening up this language to clarify what can directly concluded. (This has been done pretty carefully already, but there are a few places, e.g. the 2 places where fluidity is defined) My recommendation would be a word or phrase that indicates the observations reflect the ordering of the headgroup-chain interface, or such as headgroup packing, or something similar (which is language used elsewhere in the text).

We thank the reviewer for noticing this, and we agree with the suggested clarifications. We have now modified the text when defining fluidity, to make this point clearer. The changes have been highlighted in blue:

p. 10 now reads: “Here, we use the term “fluidity” referring to the order of the membrane headgroup-chain interface for a phospholipid bilayer... The direct relationship between the LAURDAN spectral properties and the order of the glycerol backbone interface was recently confirmed with nuclear magnetic resonance spectroscopy (H-NMR).”

p. 22 now reads: “When using the term fluidity obtained from LAURDAN fluorescence, we refer to changes in the order of the headgroup-chain interface, considering any process that can alter lipid rotational or translational rates.”